

**Atmospheric Band Fitting Coefficients Derived from Self-Consistent Rocket-Borne**

2                                      **Experiment**

Mykhaylo Grygalashvyly[1], Martin Eberhart[5], Jonas Hedin[4], Boris Strelnikov[1], Franz-Josef
Lübken[1], Markus Rapp[1,2], Stefan Löhle[5], Stefanos Fasoulas[5], Mikhail Khaplanov[4†],  Jörg

5                          Gumbel[4], and Ekaterina Vorobeva[3]

[1]Leibniz-Institute of Atmospheric Physics at the University Rostock in Kühlungsborn,
Schloss-Str. 6, D-18225 Ostseebad Kühlungsborn, Germany
[2]Deutsches Zentrum für Luft- und Raumfahrt, Institut für Physik der Atmosphäre,
Oberpfaffenhofen, Germany
[3]Department of Atmospheric Physics, Saint-Petersburg State University, Universitetskaya
Emb. 7/9, 199034, Saint-Petersburg, Russia
[4]Department of Meteorology (MISU), Stockholm University, Stockholm, Sweden
[5]University of Stuttgart, Institute of Space Systems, Stuttgart, Germany
[†]Deceased
**Abstract**
Based on self-consistent rocket-borne measurements of temperature, densities of atomic
oxygen and neutral air, and volume emission of the Atmospheric Band (762 nm) we
examined the one-step and two-step excitation mechanism of $O_2(b^1\Sigma_g^+)$ for night-time
conditions. Following McDade et al. (1986), we derived the empirical fitting coefficients,
which parameterize the Atmospheric Band emission $O_2(b^1\Sigma_g^+ - X^3\Sigma_g^-)(0,0)$ in terms of the
atomic oxygen concentrations. This allows to derive atomic oxygen concentration from night-
time observations of Atmospheric Band emission $O_2(b^1\Sigma_g^+ - X^3\Sigma_g^-)(0,0)$. The derived
empirical parameters can also be utilised for Atmospheric Band modelling. Additionally, we
derived fit function and corresponding coefficients for combined (one- and two-step)



mechanism. Simultaneous and true common volume measurements of all the parameters used
in this derivation, i.e. temperature and density of the background air, atomic oxygen density,
and volume emission rate, is the novelty and the advantage of this work.
**1. Introduction**
The mesopause region is essential to understand the chemical and physical processes in the
upper atmosphere, as well as coupling between atmospheric layers. This region is
characterised by different airglow emissions and, particularly, by emissions in the
Atmospheric Band that form the excited state of molecular oxygen $O_2(b^1\Sigma_g^+)$. Airglow
observation in the Atmospheric Band is a useful method to study dynamical processes in the
mesopause region. There have been a number of reports of gravity waves (GWs) detection in
this band (Noxon, 1978; Viereck and Deehr, 1989; Zhang et al., 1993). Planetary wave
climatology has been investigated by the Spectral Airglow Temperature Imager (SATI)
instrument (Lopez-Gonzales et al., 2009). In addition, the tides parameters have been reported
from SATI (Lopez-Gonzales et al., 2005) and High Resolution Doppler Imager (HRDI)
observations (Marsh et al., 1999). In numerous works Takahashi inferred the temperature by
Atmospheric Band observation (Takahashi et al., 1990; 1992; 2011). Furthermore, the
response of mesopause temperature and atomic oxygen during major sudden stratospheric
warming was studied utilising Atmospheric Band emission by Shepherd et al. (2010). Various
works have focused on Atmospheric Band emission modelling with respect to gravity waves
and tides (e.g. Hickey et al., 1993; Leko et al., 2002; Liu and Swenson, 2003). The specific
theory of the gravity wave effects on $O_2(b^1\Sigma_g^+)$ emission was derived in Tarasick and
Shepherd (1992). Moreover, Atmospheric Band observations have been widely utilised to
infer atomic oxygen, which is an essential chemical constituent for energetic balance in the
extended mesopause region (e.g. Hedin et al., 2009, and references there in), and ozone





concentration (Mlynczak et al., 2001). Although there is a large field of application of
Atmospheric Band emissions, there is a lack of knowledge on processes of the $O_2\left(b^1\Sigma_g^+\right)$
population. Two main mechanisms of population were proposed: the first is the direct
population from three-body recombination of atomic oxygen (e. g. Deans et al., 1976); the
second is the so-called two-step mechanism, which assumes an intermediate excited precursor
$O_2^*$ (e. g. Witt et al., 1984; Greer et al., 1981). It has been shown by laboratory experiments
that the first mechanism alone has not explained observed emissions (Young and Sharpless,
1963; Clyne at al., 1965; Young and Black, 1966; Bates, 1988). The second mechanism
entails a discussion about the precursor excited state and additional ambiguities in their
parameters (e.g. Greer et al., 1981; Ogryzlo et al., 1984). Thus, Witt et al. (1984) proposed
hypothesis that the $O_2(c^1\Sigma_u^-)$ state is, possibly, precursor; López-González et al. (1992a)
suppose that the precursor could be $O_2(^5\Pi_g)$; Wildt et al. (1991) found by laboratory
measurements that it could be $O_2(A^3\Sigma_u^+)$. Hence, the problem of identification is still not
solved. The breakthrough has been done after the ETON 2 (Energy Transfer in the Oxygen
Nightglow) rocket experiment. ETON 2 mission yielded empirical fitting parameters that
allow to quantify the $O_2\left(b^1\Sigma_g^+\right)$ (and, consequently, volume emission) by known O or atomic
oxygen by known volume emission (McDade et al., 1986). Despite the significance of this
work, there was an ambiguity in the derived fitting parameters because the temperature and
density of air (necessary for derivation) were taken from CIRA-72 and MSIS-83 models,
which leads to the loss of self-consistency (e.g. Murtagh et al., 1990) and, consequently, to
essential biases. Thus, more solid knowledge on these fitting coefficients based on consistent
measurements of atomic oxygen, volume emission of Atmospheric Band, and temperature
and density of background atmosphere is desirable. In this paper we present real common
volume in-situ measurements of these parameters performed in the course of WADIS-2
sounding rocket mission. In the next chapter, we describe the rocket experiment and obtained



data relevant for our study. In chapter 3, to make the paper easier to understand, we repeat
some theoretical approximations from McDade et al. (1986). The obtained results of our
calculations are discussed in chapter 4. Concluding remarks and summary are given in the last
chapter.

## 2. Rocket experiment Description


The WADIS (Wave propagation and dissipation in the middle atmosphere: Energy budget and
distribution of trace constituents) sounding rocket mission aimed to simultaneously study the
propagation and dissipation of GWs and measure the concentration of atomic oxygen. It
comprised two field campaigns conducted at the Andøya Space Center (ASC) in northern
Norway (69°N, 16°E). The WADIS-2 sounding rocket was launched during the second
campaign on 5 March 2015 at 01:44:00 UTC, that is, night-time conditions. For a more
detailed mission description, the reader is referred to Strelnikov et al. (2017) and the
accompanying paper by Strelnikov et al. (2018).
The WADIS-2 sounding rocket was equipped with the CONE instrument to measure absolute
neutral air density and temperature with high spatial resolution, instrument for atomic oxygen
density measurements FIPEX (Flux Probe Experiment) and the Airglow Photometer for
atmospheric band (762 nm) volume emission observation.
CONE (COmbined measurement of Neutrals and Electrons), operated by IAP (Leibniz
Institute of Atmospheric Physics at the Rostock University), is a classical triode type
ionisation gauge optimised for a pressure range between $10^{-5}$ to 1 mbar. The triode system is
surrounded by two electrodes: Whilst the outermost grid is biased to +3 to +6 V to measure
electron densities at a high spatial resolution, the next inner grid (-15 V) is meant to shield the
ionisation gauge from ionospheric plasma. CONE is suitable for measureing absolute neutral
air number densities at altitude range between 70 and 120 km. To obtain absolute densities,



the gauges are calibrated in the laboratory using a high-quality pressure sensor, like a
Baratron. The measured density profile can be further converted to a temperature profile
assuming hydrostatic equilibrium. For a detailed description of the CONE instrument, see
Giebeler et al. (1993) and Strelnikov et al. (2013).
The Airglow Photometer operated by MISU (Stockholm University, Department of
Meteorology) measures the emission of the molecular oxygen Atmospheric Band around 762
nm from the overhead column, from which volume emission rate is inferred by
differentiation. A suitable description and review of this measurement technique is given by
Hedin et al. (2009).
The aim of the FIPEX developed by the IRS (Institute of Space Systems, University of
Stuttgart) is to measure the atomic oxygen density along the rocket trajectory with high spatial
resolution. It employs two types of solid electrolyte sensors that differ in their electrode
material. Platinum electrodes are sensitive to both molecular and atomic oxygen, whilst gold
electrodes show a selective sensitivity to atomic oxygen. A low voltage is applied between
anode and cathode pumping oxygen ions through the electrolyte ceramic (yttria stabilised
zirconia). The current measured is proportional to the oxygen density. Sampling is realised
with a frequency of 100 Hz and enables a spatial resolution of ~10 m. Laboratory calibrations
were done for molecular and atomic oxygen. For a detailed description of the FIPEX
instruments and their calibration techniques see Eberhart et al. (2015, 2018).

**3. Theory**

Here, we are repeating the theory of $O_2\left(b^1\Sigma_g^+ - X^3\Sigma_g^-\right)(0,0)$ night-time emissions following
McDade et al. (1986) to make our paper more readable, saving all nomenclature as in the
original paper. All utilised reactions are listed in Table 1, together with corresponding
reaction rates, branching ratios, quenching rates and spontaneous emission coefficients. Some





components have been updated according to modern knowledge, thus, deviating from the
work of McDade et al. (1986).
Assuming direct one-step mechanism as a main one for population of $O_2\left(b^1\Sigma_g^+\right)$, we can write
its concentration as a ratio of production to the loss term:

$$\left[O_2\left(b^1\Sigma_g^+\right)\right] = \frac{\varepsilon k_1[O]^2 M}{A_2 + k_2^{O_2}[O_2] + k_2^{N_2}[N_2] + k_2^{O}[O]} \ , \tag{1}$$

where $k_1$ – reaction rate for three-body recombination of atomic oxygen, $\varepsilon$ is the
corresponding fraction of recombination, $A_2$ represents the spontaneous emission coefficient,
and $k_2^{O_2}, k_2^{N_2}, k_2^{O}$ are the quenching coefficients for reactions with $O_2$, $N_2$ and O, respectively.
The volume emission for transition $O_2\left(b^1\Sigma_g^+ - X^3\Sigma_g^-\right)(0,0)$ is a product of the concentration
of excited molecules and the spontaneous emission coefficient for the given transition:

$$V_{at} = A_1\left[O_2\left(b^1\Sigma_g^+\right)\right] = \frac{A_1 \varepsilon k_1[O]^2 M}{A_2 + k_2^{O_2}[O_2] + k_2^{N_2}[N_2] + k_2^{O}[O]} \ , \tag{2}$$

where $A_1$ is spontaneous emission for reaction R5 (hereafter, nomenclature RX means the
reaction X from Table 1). In case of known temperature, volume emission and concentrations
of O, $O_2$, $N_2$, and M, the fraction of recombination can be calculated as follows:

$$\varepsilon = V_{at} \frac{A_2 + k_2^{O_2}[O_2] + k_2^{N_2}[N_2] + k_2^{O}[O]}{A_1 k_1[O]^2 M} \ . \tag{3}$$

In the case of the two-step mechanism, the unknown excited state $O_2^*$ is populated at the first
step from the reaction R7. Then, it can be deactivated by quenching (R9), spontaneous
emission (R10) or producing $O_2\left(b^1\Sigma_g^+\right)$ by the reaction R8. The concentration of these excited
molecules is given by the following expression:

$$[O_2^*] = \frac{\alpha k_1[O]^2 M}{A_3 + k_3^{O_2}[O_2] + k_3^{N_2}[N_2] + k_3^{O}[O]} \ , \tag{4}$$

where fraction of recombination $\alpha$, spontaneous emission coefficient $A_3$, quenching rates
$k_3^{O_2}, k_3^{N_2}, k_3^{O}$ – are unknown values, as well as the precursor excited state.





In the second step, $O_2^*$ is transformed into $O_2\left(b^1\Sigma_g^+\right)$, which, in turn, can be deactivated by
quenching (R2-R4) and by spontaneous emission (R6). Its concentration in the case of the
two-step mechanism is:

$$\left[O_2\left(b^1\Sigma_g^+\right)\right] = \frac{\gamma k_3^{O_2}[O_2][O_2^*]}{A_2 + k_2^{O_2}[O_2] + k_2^{N_2}[N_2] + k_2^{O}[O]} \ . \tag{5}$$

The volume emission in the case of $O_2\left(b^1\Sigma_g^+ - X^3\Sigma_g^-\right)(0,0)$ is:

$$V_{at} = \frac{A_1 \alpha k_1[O]^2 M \gamma k_3^{O_2}[O_2]}{\left(A_2 + k_2^{O_2}[O_2] + k_2^{N_2}[N_2] + k_2^{O}[O]\right)\left(A_3 + k_3^{O_2}[O_2] + k_3^{N_2}[N_2] + k_3^{O}[O]\right)} \ . \tag{6}$$

Collecting all known values on the right-hand side (RHS) and all unknown summands on the
left-hand side (LHS), equation (6) can be rearranged as follows:

$$\frac{A_3 + k_3^{O_2}[O_2] + k_3^{N_2}[N_2] + k_3^{O}[O]}{\alpha \gamma k_3^{O_2}} = \frac{A_1 k_1[O]^2 M[O_2]}{V_{at}\left(A_2 + k_2^{O_2}[O_2] + k_2^{N_2}[N_2] + k_2^{O}[O]\right)} \ . \tag{7}$$

Omitting emissive summand $A_3$ as non-effective loss (McDade et al., 1986) we can transform
(7) into the following expression:

$$C^{O_2}[O_2] + C^{O}[O] = \frac{A_1 k_1[O]^2 M[O_2]}{V_{at}\left(A_2 + k_2^{O_2}[O_2] + k_2^{N_2}[N_2] + k_2^{O}[O]\right)} \ , \tag{8}$$

where $C^{O_2} = \left(1 + k_3^{N_2}[N_2]/k_3^{O_2}[O_2]\right)/\alpha\gamma$ and $C^{O} = k_3^{O}/\alpha\gamma k_3^{O_2}$ are the fitting coefficients
that can be calculated by the least square fit (LSF) procedure. We calculated them based on
our measurements and will discuss the results in the following chapter.

**4. Results and Discussion**

Figure 1 shows input data for our calculations: temperature from CONE instrument (Fig. 1a),
number density of air (Fig. 1b), atomic oxygen concentration measured by FIPEX (Fig. 1c)
and volume emission at 762 nm from photometric instrument (Fig. 1d). A temperature
minimum of ~158 K was observed at 104.2 km. A local temperature peak was measured at





98.9 km with values of 204.5 K. The secondary temperature minimum was visible at 95.4 km
and amounted to ~173 K. Atomic oxygen concentration (Fig. 1c) had a peak of ~$4.7 \cdot 10^{11}$ [cm$^-$
$^3$] at 97.2 km and approximately coincided with the secondary temperature peak. The peak of
volume emission was detected between 95 and 97 km with values of more than 1700
[phot.·cm$^{-3}$·s$^{-1}$]; this is slightly beneath the atomic oxygen corresponding maximum and
slightly above the secondary temperature minimum. Note, this point to the competition of the
temperature and the atomic oxygen concentration in the processes of atomic oxygen excited
state  $O_2\left(b^1\Sigma_g^+\right)$ formation. Independently of the mechanism of atmospheric band emission
(Eq. 2 or Eq. 6), the numerator is directly proportional to the square of atomic oxygen
concentration and inversely proportional to the third power of the temperature (via reaction
rate $k_1$ and $M$, considering the ideal gas low). Our rocket experiment shows an essential
difference of emissions between ascending and descending flights (see Strelnikov et al.,
2018). It also demonstrates a significant variability in other measured parameters, including
neutral temperature and density as well as atomic oxygen density. This suggests that, in the
case of the ETON 2 experiments, the temporal extrapolation of atomic oxygen for the time of
the emission measurement flight (which was approximately 20 min earlier) may lead to
serious biases in estimations because, as one can see from Eq. 2 and Eq. 6, volume emission
depends on the atomic oxygen concentration quadratically. Since the best quality data were
obtained during the descent of the WADIS-2 rocket flight, we chose this data set for our
analysis (Strelnikov et al., 2018). The region above 104 km is subject to auroral
contamination. In the region below 92 km, negative values may occur in the volume emission
profile as the result of self-absorption in the denser atmosphere below the emission layer.
Hence, we considered the region near the peak of emission between 92 km and 104 km as
most appropriate for our study. The comparisons of our measurements with other
observations, as well as with the results of modelling are presented in several papers (e.g.
Eberhart et al., 2018; Strelnikov et al., 2018).




## 4.1 One-step mechanism


Figure 2 shows the fraction of recombination $\varepsilon$ calculated according to Eq. (3), which is
necessary to form $O_2\left(b^1\Sigma_g^+\right)$ under the assumption that the direct three-body recombination of
atomic oxygen is the main mechanism. As is expected, $\varepsilon$ is scattered approximately in the
range [0.07; 0.13], which is in good agreement with the values derived by McDade et al.
(1986). The averaged value amounts to 0.1. By the physical nature of this value, the fraction
of recombination should not depend on altitude, but Fig. 2 shows the strong altitude
dependence. Such behaviour of $\varepsilon$ means that measured atmospheric emissions may not be
explained merely in light of direct excitation mechanism if $\varepsilon$ is independent of temperature.
Hence, we plot values of $\varepsilon$ depending on measured temperature in Figure 3. The values are
distributed not randomly and show clear functional dependence. This dependence has a
complex nonlinear form. The spiral shape points to the existence of a second parameter,
which is probably the pressure. The reaction rates in general cases are the functions of
temperature and pressure (e. g. Troe, 1979). Hence, $\varepsilon$, which represents the ratio of the
reaction rates of different branches, must depend, in general, on temperature and pressure.
Correct functional relation $\varepsilon = \varepsilon(T, p)$ can be obtained only through laboratory
measurements. In light of the analysis of our rocket experiment, we can only state that such
functional dependence may exist. Hence, an explanation of atmospheric band emission via
one-step mechanism is, generally speaking, possible. On the other hand, an existence of
functional dependence is the necessary but not sufficient condition to state that the one-step
mechanism populates $O_2\left(b^1\Sigma_g^+\right)$. Moreover, although the population via one-step mechanism
alone is possible, it is improbable because laboratory experiments show that the direct
excitation alone may not explain observed emissions (Young and Sharpless, 1963; Clyne at





al., 1965; Young and Black, 1966; Bates, 1988). This conclusion is partially in agreement
with the conclusion from McDade et al. (1986), which stated that the one-step excitation
mechanism is not sufficient to explain the $O_2(b^1\Sigma_g^+)$ population.
The reason of some inconsistency between our and McDade et al. (1986) formulation is that,
in the case of ETON 2 experiments it was not possible to correlate $\varepsilon$ with the real temperature
at the place and in time of rocket launch. For their analysis, mean temperature profiles were
utilised from the models CIRA-72 and MSIS-83 (Hedin, 1983), which does not reproduce any
short-time dynamical fluctuations, solar cycle conditions, etc. Hence, the investigation of
correlation between temperature and $\varepsilon$ was not possible. Therefore, just in frame of our
experiment, we may not decline that $O_2(b^1\Sigma_g^+)$ is populated via one-step or other
mechanisms, but taking into account the results of laboratory measurements Bates (1988) and
theoretical investigations Wraight (1982), which infer too low $\varepsilon$ (0.03 and 0.015, respectively)
we should conclude that one-step mechanism alone does not explain observed emissions.
Hence, in the following, we check the second energy transfer mechanism.

**4.2 Two-step mechanism**

Figure 4 depicts the altitude profile of the RHS of equation (8) and profile calculated by the
least-square fit (LSF). The fitting coefficients, $C^{O2}$ and $C^O$, resulting from this fit, are amount
to 9.8 and 2.1, respectively. In such a way defined (Eq. 8) fitting coefficients do not have a
direct physical meaning. However, they have a physical meaning in several limit cases. If the
quenching coefficients of a precursor with molecular nitrogen are much smaller than those
with molecular oxygen $\left(k_3^{N_2} \ll k_3^{O_2}\right)$, then $\alpha\gamma = 1/C^{O2}$. In our case $\alpha\gamma = 0.102$. In other
words, in the case of two-step formation of $O_2(b^1\Sigma_g^+)$ with energy transfer agent $O_2$, the total
efficiency $\alpha\gamma$ amounts to 10.2%, which is the lowest amongst known values. Based on rocket





experiment data analysis (ETON), Witt et al. (1984) obtained $\alpha\gamma = 0.12 - 0.2$. According to
McDade et al. (1986), for the case with $k_2^O = 8 \cdot 10^{-14}$, the total efficiencies are 0.15 and
0.21 for temperature profiles adopted from MSIS-83 and CIRA-72, respectively. The analyses
of López-González et al. (1992a, c), adopted $O_2$, $N_2$, and temperature profiles from the model
(Rodrigo et al., 1991), showed a total efficiency of 0.16. In contrast to our work, all
investigations mentioned above utilised the temperature and atmospheric density from models
which describe a mean state of the atmosphere. This is a possible reason for discrepancy in
the results. Total efficiency may serve as an auxiliary quantity to identify the precursor.
According to the physical meaning of efficiency, it may not be larger than 1. Hence, α, γ, as
well as the total efficiency are smaller than 1. Consequently, $\gamma = tot.eff./\alpha < 1$, and we
can examine potential candidates for $O_2^*$ with this criterion. From an energetic point of view,
only four bound states of molecular oxygen can be considered as an intermediate state for the
$O_2\left(b^1\Sigma_g^+\right)$ population: $O_2(A^3\Sigma_u^+), O_2(A'^3\Delta_u), O_2(c^1\Sigma_u^-)$, and $O_2(^5\Pi_g)$ (Greer et al., 1981;
Wraight, 1982; Witt et al., 1984; McDade et al., 1986; López-González et al., 1992c). For
better readability, we will partially repeat a table from López-González et al. (1992b, c) with
known α in our work (Table 2). From Table 2, it can be seen that only $O_2(A'^3\Delta_u)$ and
$O_2(^5\Pi_g)$ fit to the criterion of $\gamma = 0.102/\alpha < 1$.
The second expression that helps to clarify the choice of the precursor is the ratio of
quenching rates. In the limit of low quenching with molecular nitrogen $\left(k_3^{N_2} \ll k_3^{O_2}\right)$, the ratio
of fitting coefficients equals the ratio of the quenching rates of atomic and molecular oxygens
$\left(C^O/C^{O_2} = k_3^O/k_3^{O_2}\right)$. An analysis from the ETON 2 rocket experiment yields values of
quenching coefficients ratios of potential precursor are 3.1 and 2.9 for temperatures from
CIRA-72 and MSIS-83, respectively. This is close to the value from Ogryzlo et al. (1984),
who found $k_3^O/k_3^{O_2} = 2.6$ by laboratory measurements; however, as was noted in their work,
substitution of these values into the equation for emission yields 16 % of the observed



emission (Ogryzlo et al., 1984). These findings point to the possibility of a too high measured
ratio $k_3^O/k_3^{O_2}$ as the result of too strong quenching of precursor by atomic oxygen. Our value
of quenching ratios $k_3^O/k_3^{O_2}$ amounts to 0.21. There is not enough information on measured
values for bound states of molecular oxygen. Laboratory measurements for $O_2(A^3\Sigma_u^+)(v =$
$0 - 4)$, $O_2(A^3\Sigma_u^+)(v = 2)$, and $O_2(c^1\Sigma_u^-)$ infer the values of $k_3^O/k_3^{O_2}$ ratio to be 30±30,
100±15, and 200±20, respectively (Kenner and Ogryzlo, 1980; Kenner and Ogryzlo, 1983a,
1983b; Kenner and Ogryzlo, 1984). On the other hand, Slanger et al. (1984) found a lower
limit of $O_2(A^3\Sigma_u^+)(v = 8)$ quenching by $O_2$ must be ≥8·10$^{-11}$. If the results from Slanger et al.
(1984) were applied to the results from Kenner and Ogryzlo (1980, 1984) for $k_3^{O_2}$, then the
ratio of $k_3^O/k_3^{O_2}$ would be two orders lower. This short discussion illustrates a strong scattering
of this ratio. For our two potential candidates ($O_2(A'^3\Delta_u)$ and $O_2(^5\Pi_g)$), there is information
about $k_3^O/k_3^{O_2}$ ratio for only $O_2(A'^3\Delta_u)$. Through the comprehensive analysis of known rocket
experiments, López-González et al. (1992a, b, c) inferred that the upper limit of the ratio
amounts to 1. Hence, our value of $k_3^O/k_3^{O_2} = 0.21$ agrees with this result. Consistent
information from laboratory experiments on the ratio for $O_2(^5\Pi_g)$ is absent. Thus, we can
propose as potential candidates for precursor both $O_2(A'^3\Delta_u)$ and $O_2(^5\Pi_g)$; however, we are
not able to identify which of these two is more preferable.
In order to illustrate the application of the newly derived fitting coefficients we show Figure 5
with atomic oxygen concentration from FIPEX (black line), from NRL MSISE-00 reference
atmosphere model (Picone et al., 2002) (red line); calculated with McDade et al. (1986)
coefficients (blue line), and with our fitting coefficients for the two-step mechanism (green
line). In the region 90-98 km, i.e. beneath atomic oxygen peak (see Fig. 1d) fitting
coefficients from this paper better then McDade coefficients (MSIS-83 case). Our fitting
coefficients and fitting coefficients of McDade give similar approximation above atomic




oxygen peak (~98-104 km). The atomic oxygen retrieved with our fitting coefficients
satisfactorily reproduces measurements.
**4.3 Combined mechanism**

In the most general case, the $O_2\left(b^1\Sigma_g^+\right)$ population passes through two channels: directly and
via precursor. In fact, theoretical calculations from Wraight (1982) and laboratory
measurements from Bates (1988) predicted a direct population with efficiencies of 0.015 and
0.03, respectively, which is not sufficient to explain the observed emissions (Bates, 1988,
Greer et al., 1981; Krasnopolsky, 1986). A similar value, ε=0.02, was shown in the analysis
by López-González et al. (1992b, c). We investigated a combined mechanism based on the
LSF     calculation     and     fit     function     (derivation     in     Appendix):

$$\frac{[O_2] + D_1[O]}{D_2 + \tilde{\varepsilon}(1 + D_1\,[O]/[O_2])} = \frac{A_1 k_1 [O]^2 M[O_2]}{V_{at}\left(A_2 + k_2^{O_2}[O_2] + k_2^{N_2}[N_2] + k_2^{O}[O]\right)} \, , \qquad (9)$$

where, hereafter, tildes denote that these are values for combined mechanism and do not equal
to the values for one-step or two-step mechanisms (Ch. 4.1 and 4.2); $D_1 = \tilde{k}_3^O/\tilde{k}_3^{O_2}$ and
$D_2 = \tilde{\alpha}\tilde{\gamma}$ are the fitting coefficients, which refer to the ratio of quenching rates and total
efficiency for two-step channel, respectively. The fitting coefficients were calculated for two
limit cases $\tilde{\varepsilon}$=0.015 (Wraight, 1982), $\tilde{\varepsilon}$=0.03 (Bates, 1988) and for the averaged case $\tilde{\varepsilon}$=0.022.
They are listed in Table 3. The altitude profile of the RHS of equation (9) and calculated fit-
function are plotted in Figure 6. The deviations of fit function between limits and averaged
values are negligible, hence, we only show the averaged case. Thus, we can recommend for
future investigations the values of averaged case (last column of Tab. 3). Analogously to the
two-step mechanism (Ch. 4.2), for the case of combined mechanism $\tilde{\gamma} = tot.eff./\tilde{\alpha} < 1$.
Taking into account the lowest value for total efficiency, the precursor should satisfy $\tilde{\alpha} >$
0.073. Consequently, only $O_2(A'^3\Delta_u)$ and $O_2(^5\Pi_g)$ satisfy this criterion (see Tab 2). The
upper limit of the ratio $k_3^O/k_3^{O_2} < 1$ for $O_2(A'^3\Delta_u)$, derived by López-González et al. (1992a,





b, c), is in agreement with our calculations (~0.2-0.4). As it is noted above, the ratio for
$O_2(^5\Pi_g)$ is unknown.
Figure 7 illustrates volume emissions derived (black lines) with fitting coefficients of
McDade et al. (1986) for MSIS-83 (Fig. 7c) case and CIRA-72 case (Fig. 7d), and with our
newly derived fitting coefficients for two-step (Fig. 7a) and combined ($\tilde{\varepsilon} = 0.022$)
mechanisms (Fig. 7b) in comparison with measured one (red lines). All of derived volume
emission profiles (black lines) were calculated based on the the temperature, concentration of
surrounding air, and concentration of atomic oxygen from our rocket launch. The calculations
with combined mechanism (Eq. 9) and two-step energy transfer mechanism (Eq. 8) give
almost identical results. The results obtained with new fitting coefficients are in satisfactory
agreement with the measured volume emissions at the peak and above, whereas the McDade
coefficients related to the temperature from CIRA-72 give better approximations below the
volume emission peak (92 km). The coefficients of McDade related to the temperature from
MSIS-83 are in better agreement with our results and are almost identical above the volume
emission peak. Note that both mechanisms with newly derived coefficients give three of five
points in the vicinity of uncertainties of measured values (see Fig. 7a and Fig 7b), whereas
McDade coefficients for MSIS-83 case (Fig. 7c) and for CIRA-72 (Fig. 7d) give just one and
two point, respectively. Hence, we can recommend our newly derived coefficients either for a
two-step energy transfer process or for combined mechanism. We consider combined
mechanism more preferable as it is more general.

**5. Summary and conclusions**

Based on the rocket-born true common volume observations of atomic oxygen, atmospheric
band emission (762 nm), and density and temperature of the background atmosphere, the one-
step, two-step and combined mechanisms of $O_2\left(b^1\Sigma_g^+\right)$ formation were analysed. Our





calculations show that in the case of the one-step mechanism, the fraction of atomic oxygen
recombination ε depends on altitude. The one-step mechanism inferred the functional
dependence of ε on temperature. It has a nonlinear character because the fraction of
recombination ε, in general, depends on temperature and pressure. Nevertheless, we consider
one-step direct excitation as less probable for the reasons discussed above (Ch.4.1). In the
context of the rocket experiment, we do not have a possibility to retrieve functional
dependence $\varepsilon = \varepsilon(T, p)$, which poses a task for future laboratory measurements.
For the case of the two-step mechanism, we found new coefficients for fit function in
accordance with McDade et al. (1986), based on self-consistent temperature, atomic oxygen
and volume emission observation. These coefficients amounted to $C^{O2}$=9.8 and $C^O$=2.1. The
general implication of these results is parameterisation of volume emission in terms of known
atomic oxygen. This can be utilised either for atmospheric band volume emission modelling
or for estimation of atomic oxygen by known volume emission. We identified two candidates
for the intermediate state of $O_2^*$. Our results show that $O_2(A'^3\Delta_u)$ or $O_2(^5\Pi_g)$ may serve as a
precursor.

355        Taking into account both channels of $O_2\left(b^1\Sigma_g^+\right)$ formation, we proposed a combined

mechanism. In this case, atomic oxygen via volume emission or volume emission based on
known atomic oxygen can be calculated by equation (9). Recommended fitting coefficients
amounted to $D_1$=0.231 and $D_2$=0.08, with the efficiency of the direct channel as
$\tilde{\varepsilon} = 0.022$. These coefficients have a sense of total efficiency $(\tilde{\alpha}\tilde{\gamma})$ and a ratio of quenching
coefficients $(\tilde{k}_3^O/\tilde{k}_3^{O_2})$ for the two-step channel. The analysis of their values indicates that
$O_2(A'^3\Delta_u)$ and $O_2(^5\Pi_g)$ may serve as possible precursors for the two-step channel.
Unfortunately, in the context of our rocket experiment, we do not have the possibility to
figure out which mechanism is true. Nevertheless, we consider the combined mechanism as
more relevant to nature, because it has a higher generality. This conclusion does not





contradict to the current point of view that the two-step mechanism is dominant because $\tilde{\varepsilon}$ is
assumed to be 1.5-3 %. Moreover, it is possible that in the reality the mechanism much more
complex and it has multi-channel or more than two-step nature.

**Appendix.**

We consider photochemical equilibrium for the night-time $O_2\left(b^1\Sigma_g^+\right)$ concentration. If
$O_2\left(b^1\Sigma_g^+\right)$ is produced via both channels, the equilibrium concentration is given by the
following expression:

$$\left[O_2\left(b^1\Sigma_g^+\right)\right] = \frac{\tilde{\varepsilon}k_1[O]^2M + \tilde{\gamma}\tilde{k}_3^{O_2}[O_2][O_2^*]}{A_2 + k_2^{O_2}[O_2] + k_2^{N_2}[N_2] + k_2^{O}[O]} \,, \tag{A1}$$

where the tilde denotes the combined mechanism, $A_1, k_1, k_2^{O_2}, k_2^{N_2}, k_2^{O}, \tilde{k}_3^{O_2}$ are the ratios for
corresponding processes (see Tab. 1) and $O_2^*$ is the unknown precursor.
Considering this precursor in photochemical equilibrium, we can obtain the following
expression for its concentration:

$$[O_2^*] = \frac{\tilde{\alpha}k_1[O]^2M}{\tilde{A}_3 + \tilde{k}_3^{O_2}[O_2] + \tilde{k}_3^{N_2}[N_2] + \tilde{k}_3^{O}[O]} \,, \tag{A2}$$

where efficiency $\tilde{\alpha}$, $\tilde{A}_3$ is the unknown spontaneous emission coefficient of $O_2^*$ and
$\tilde{k}_3^{O_2}, \tilde{k}_3^{N_2}, \tilde{k}_3^{O}$ are the unknown quenching rates for $O_2^*$.
Substituting A2 into A1 and into expression for volume emission we obtain:
$V_{at} = A_1\left[O_2\left(b^1\Sigma_g^+\right)\right] =$

$$= \frac{A_1 k_1 [O]^2 [O_2] M}{A_2 + k_2^{O_2}[O_2] + k_2^{N_2}[N_2] + k_2^{O}[O]} \left( \frac{\tilde{\varepsilon}}{[O_2]} + \frac{\tilde{\alpha}\tilde{\gamma}\tilde{k}_3^{O_2}}{\tilde{A}_3 + \tilde{k}_3^{O_2}[O_2] + \tilde{k}_3^{N_2}[N_2] + \tilde{k}_3^{O}[O]} \right). \tag{A3}$$

We assume that, in analogy with two-step mechanism, a spontaneous emission $\widetilde{A_3}$ of $O_2^*$ is
much smaller than the quenching, and we utilised traditional assumption about low quenching





with molecular nitrogen $\left(\tilde{k}_3^{N_2} \ll \tilde{k}_3^{O_2}\right)$, which is commonly used to analyse a potential
precursor. In this case, A3 can be rearranged as follows:

$$\frac{[O_2] + \dfrac{\tilde{k}_3^O}{\tilde{k}_3^{O_2}}[O]}{\tilde{\alpha}\tilde{\gamma} + \tilde{\varepsilon}\left(1 + \dfrac{\tilde{k}_3^O}{\tilde{k}_3^{O_2}}[O]/[O_2]\right)} = \frac{A_1 k_1 [O]^2 M[O_2]}{V_{at}\left(A_2 + k_2^{O_2}[O_2] + k_2^{N_2}[N_2] + k_2^O[O]\right)} . \qquad (A4)$$

We defined unknown fitting coefficients $D_1 \equiv \tilde{k}_3^O / \tilde{k}_3^{O_2}$ and $D_2 \equiv \tilde{\alpha}\tilde{\gamma}$. Expression A4 was
utilised to calculate them with LSF.

**Acknowledgements.**

The authors are thankful to Prof. Dr. V. A. Yankovsky, Prof. Dr. W. Ward, and PD Dr. G. R.
Sonnemann for helpful suggestions and useful discussions. This work was supported by the
German Space Agency (DLR) under grant 50 OE 1001 (project WADIS). The authors thank
DLR-MORABA for their excellent contribution to the project by developing the complicated
WADIS payload and campaign support together with the Andøya Space Center, as well as H.-
J. Heckl and T. Köpnick for building the rocket instrumentation.
The rocket-borne measurements and calculated data shown in this paper are available via
IAP's ftp server at ftp://ftp.iap-kborn.de/data-in-publications/GrygalashvylyACP2018.

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





**Table 1**. List of reactions with corresponding reaction rates (for three-body reactions [cm$^6$
molecule$^{-2}$ s$^{-1}$] and for two-body reactions [cm$^3$ molecule$^{-1}$ s$^{-1}$]), quenching coefficients, and
spontaneous emission coefficients (s$^{-1}$) used in the paper.

|  | Reaction | Coefficient | Reference |
|---|---|---|---|
| R1 | $O + O + M \xrightarrow{\varepsilon k_1} O_2(b^1\Sigma_g^+) + M$ | $k_1 = 4.7 \cdot 10^{-33}(300/T)^2$ $\varepsilon - unknown$ | Campbel and Gray (1973) |
| R2 | $O_2(b^1\Sigma_g^+) + O_2 \xrightarrow{k_2^{O_2}} products$ | $k_2^{O_2}$ $= 7.4 \cdot 10^{-17}T^{0.5}e^{-\frac{1104.7}{T}}$ | Zagidullin et al. (2017) |
| R3 | $O_2(b^1\Sigma_g^+) + N_2 \xrightarrow{k_2^{N_2}} products$ | $k_2^{N_2} = 8 \cdot 10^{-20}T^{1.5}e^{\frac{503}{T}}$ | Zagidullin et al. (2017) |
| R4 | $O_2(b^1\Sigma_g^+) + O \xrightarrow{k_2^{O}} products$ | $k_2^{O} = 8 \cdot 10^{-14}$ | Slanger and Black (1979) |
| R5 | $O_2(b^1\Sigma_g^+) \xrightarrow{A_1} O_2 + h\nu(762nm)$ | $A_1 = 0.0834$ | Newnham and Ballard (1998) |
| R6 | $O_2(b^1\Sigma_g^+) \xrightarrow{A_2} O_2 + h\nu(total)$ | $A_2 = 0.088158$ | Yankovsky et al. (2016) |
| R7 | $O + O + M \xrightarrow{\alpha k_1} O_2^* + M$ | $\alpha - unknown$ | |
| R8 | $O_2^* + O_2 \xrightarrow{\gamma k_3^{O_2}} O_2(b^1\Sigma_g^+) + O_2$ | $\gamma - unknown$ | |
| R9 | $O_2^* + O_2, N_2, O \xrightarrow{k_3^{O_2}, k_3^{N_2}, k_3^{O}} prod.$ | $k_3^{O_2}, k_3^{N_2}, k_3^{O} - unknown$ | |
| R10 | $O_2^* \xrightarrow{A_3} O_2 + h\nu$ | $A_3 - unknown$ | |


**Table 2.** Efficiencies α of the different excited states of O$_2$.

| $O_2(c^1\Sigma_u^-)$ | $O_2(A'^3\Delta_u)$ | $O_2(A^3\Sigma_u^+)$ | $O_2(^5\Pi_g)$ | Reference |
|---|---|---|---|---|
| 0.03 | 0.12 | 0.04 | 0.66 | Wraight (1982), Smith (1984) |
| 0.04 | 0.18 | 0.06 | 0.5 | Bates (1988) |
| 0.03 | 0.18 | 0.06 | 0.52 | López-González et al. (1992a, b, c) |


**Table 3.** Fitting coefficients for combined mechanism (Eq. 9) at different efficiencies.

|  | Low $\tilde{\varepsilon}$ Wraight (1982) | High $\tilde{\varepsilon}$ Bates (1988) | Averaged $\tilde{\varepsilon}$ (this work) |
|---|---|---|---|
| $\tilde{\varepsilon}$ | 0.015 | 0.03 | 0.022 |
| $D_1 = \tilde{k}_3^{O}/\tilde{k}_3^{O_2}$ | 0.211 | 0.397 | 0.231 |
| $D_2 = \tilde{\alpha}\tilde{\gamma}$ | 0.087 | 0.073 | 0.08 |






**Figures.**
Figure 1. Measurements of a) temperature (CONE), b) number density of air (CONE), c)
atomic oxygen concentration (FIPEX), d) volume emission at 762 nm (photometer).

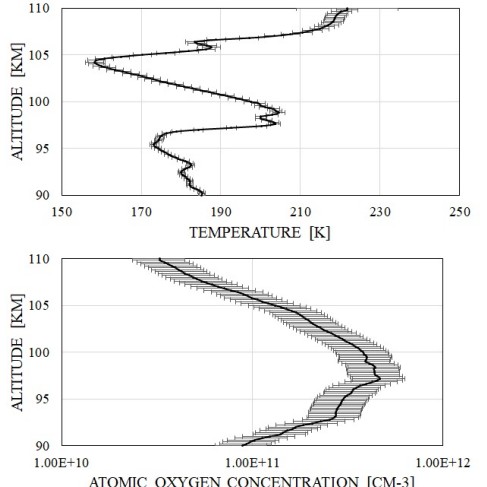
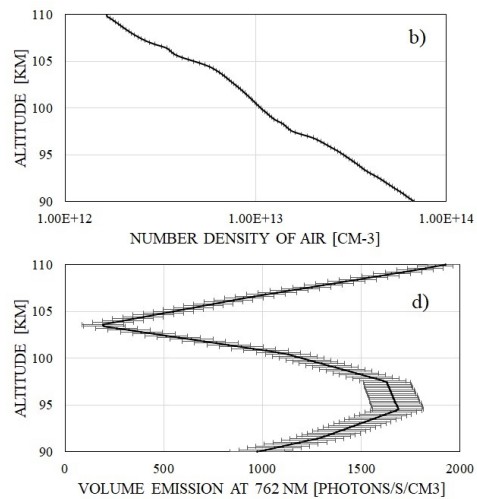




















641   Figure 2. Fraction of recombination ε for the case of one-step mechanism.

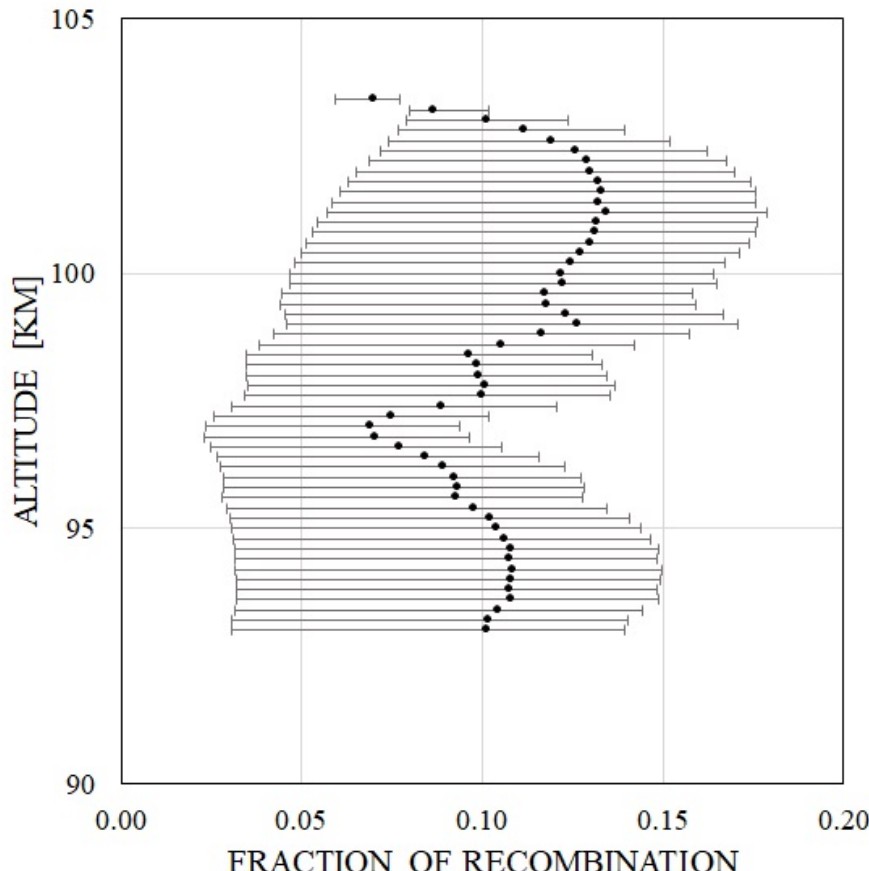















Figure 3. Correlation between fraction of recombination ε and T.

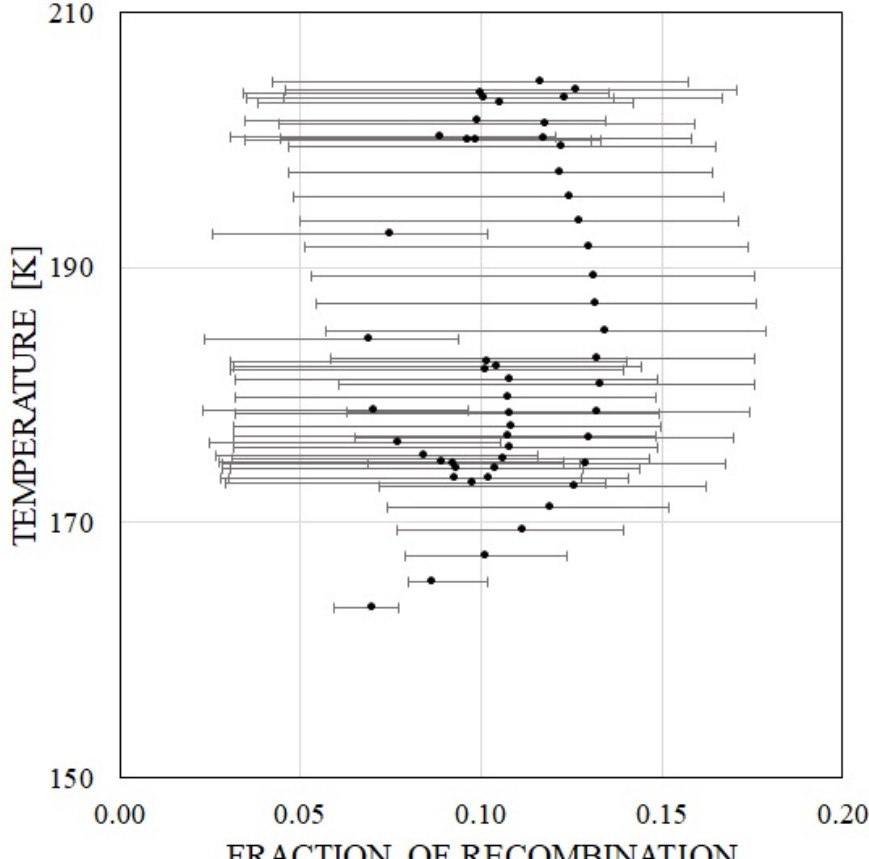















Figure 4. RHS of equation (8) and least-square fit of LHS of equation (8).

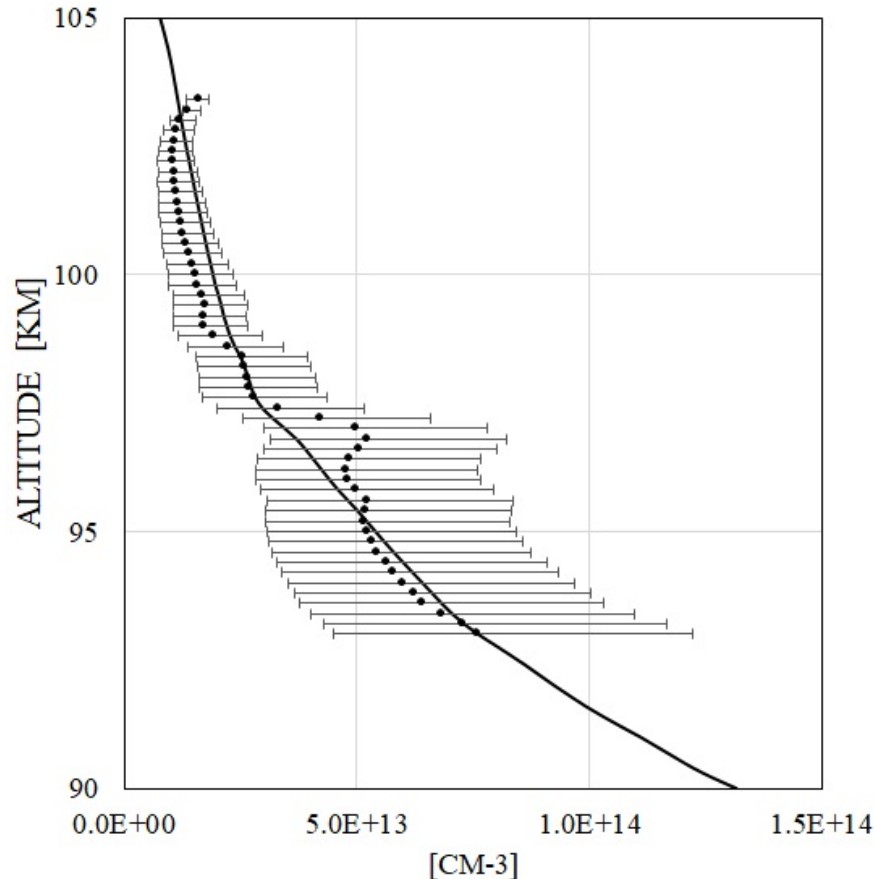















Figure 5. Atomic oxygen concentration: FIPEX (black line); model MSIS00 (red line);
derived from emission observation with McDade et al. (1986) coefficients (blue line);
calculated with newly derived fitting coefficients for the two-step mechanism (green line).

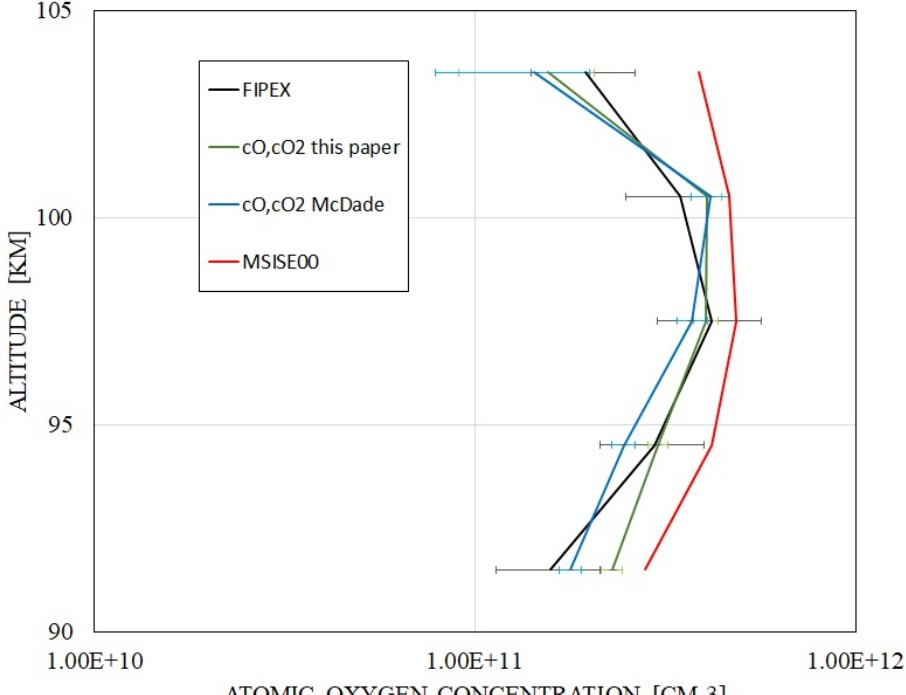














Figure 6. RHS and least-square fit of LHS of equation (9) for averaged case $\tilde{\varepsilon}$=0.0225.

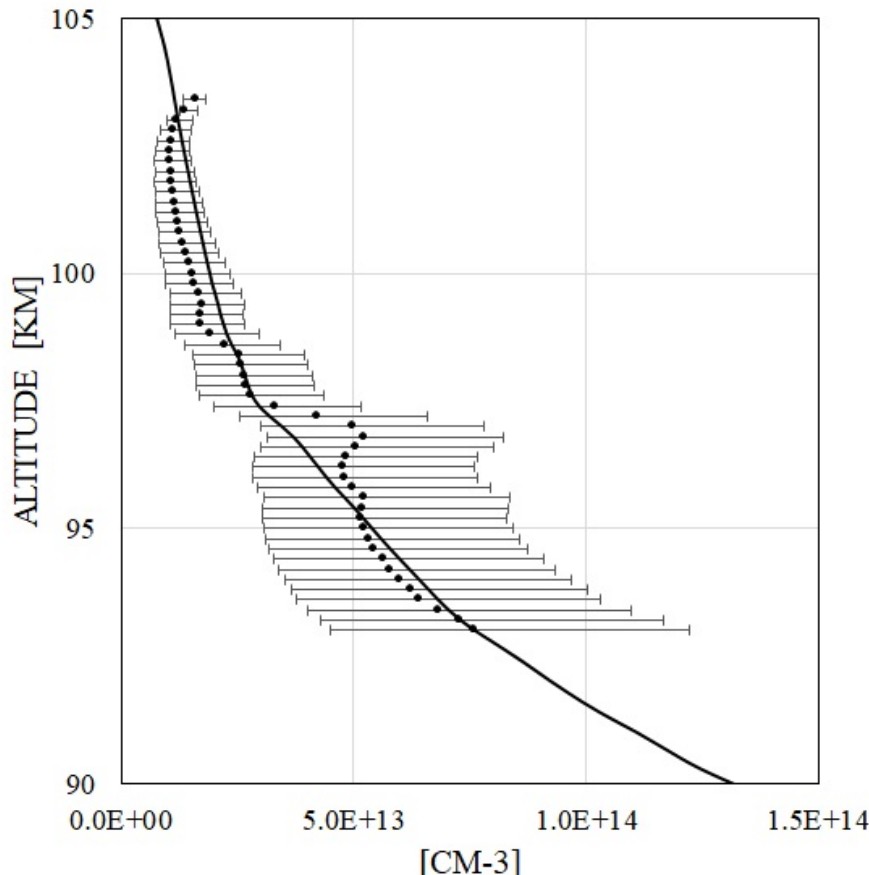














Figure 7. Volume emissions: photometer (red line); derived from atomic oxygen (black line)
with a) newly derived fitting coefficients for the two-step mechanism, b) with fitting
coefficients for combined mechanism, c) with McDade et al. (1986) coefficients, which
correspond to the MSIS-83 temperature, and with McDade et al. (1986) coefficients, which
correspond to the CIRA-72  temperatures.

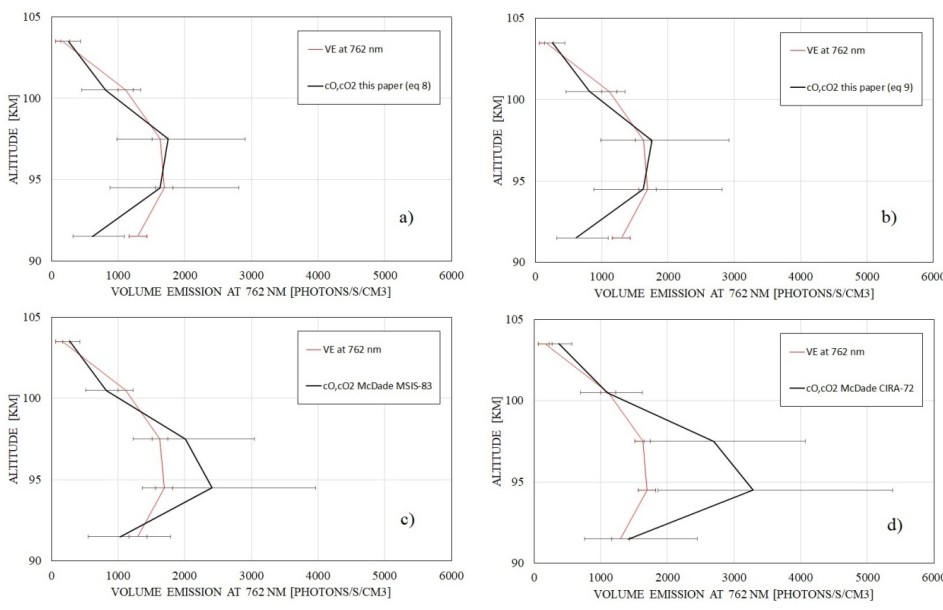
