# Peer review of "Atmospheric Band Fitting Coefficients Derived from Self-Consistent Rocket-Borne"

_Atmospheric Chemistry and Physics, 2018_

## Referee Comment (RC1) · Anonymous Referee #1 · 29 Oct 2018

Second Referee's comments on manuscript Number: acp-2018-696 Title: Atmospheric Band Fitting Coefficients Derived from Self-Consistent Rocket-Borne Experiment Author(s): Mykhaylo Grygalashvyly et al.

This paper try to confirm the chemical production processes responsible of the Atmospheric Oxygen emission observed in the Earth nightglow. Although previous works have pointed out that the excited state of molecular oxygen O2(b), responsible of this emission, is produced mainly by a transfer recombination processes, because, with the available information up to now, direct recombination of atomic oxygen process, alone, can not explain the amount of emission measured in many different experimental atmospheric airglow measurements, it is the first time that are available simultaneous measurements of the main parameters involved in the chemistry of O2(b) state: O2(b) Atmospheric emission profile, atomic oxygen, temperature, atmospheric density... and this offers the opportunity to check and improve our knowledge of the O2(b) chemistry.

This is the main goal of this study, the opportunity of using simultaneous measurements of all the parameters involved in the O2(b) chemistry to perform this investigation.

The main conclusions derived are: A total efficiency for O2(b) production of 0.10 in a transfer mechanism (close to the previous accepted values) with a quenching ratio for the precursor Ko/Ko2 of 0.21 (10-20 times smaller than the previous accepted values). These results are what have to be analysed with care and discussed. Although the authors have answered some of my previous comments, I think sometimes the authors lose the focus of work, and do many speculative work on some aspects that do not lead to any new conclusion.

First, I do not consider necessary many of the analysis devoted to atomic oxygen direct recombination process. So Figures 2 and 3, and many of the discussion refereed to them, should not be here. Once the authors obtain from direct recombination analysis that an efficiency of 0.07-0.13 is required for O2(b) production means that this process, alone, can not explain O2(b) production, following theoretical and experimental previous works (as it is said many times along the text and supported by references), then, all additional the work of considering only direct recombination process and possible dependences of the efficiency with temperature and pressure is very speculative, first because of the large value of this efficiency and second because in the altitude range of the atmospheric region considered ($\sim$10km) there is not large variation of temperature and pressure condition (although they could reach +-50K and/or a few micro bars). This is an exercise and it confuses and makes lose the focus of the work.

So Figure 2 and Figure 3 are not needed.

Figure 4, 5 and 7 need some improvements:

1) Plot some subintervals in the vertical and horizontal axes.

2) There are points (data) only at about each 3 km. Points at each 1 km should be shown, although the error bars be shown only each 3 km.

3) Figure 3 shows very good fit between about 97 and 98 km (as I can guess in figure 3 without any subdivision!). It will be interesting to show in the same figure the temperature profile (with and appropriate temperature scale in the upper horizontal axis), and the number density to see how the structure of temperature and number density can affect the fitting.

I think that the authors still should make additional work, to support these results.

I have made a few recommendations (see details that follows):

Abstract

Line 19- 20: ",we derived the empirical fitting coefficients,....(0,0) in terms of the atomic oxygen concentrations." Delete "in terms of the atomic oxygen concentrations" To read: ",we derived the empirical fitting coefficients,....(0,0)."

Line 25: "Simultaneous and true common volume measurements of all the parameters used in this derivation, i.e...." Change to: Simultaneous measurements of all the parameters involved in the theoretical calculation of the observed O2(b) emission, i.e...."

Introduction:

Line 33-35 Change: "particularly, by emissions in the Atmospheric Band that form the excited state of molecular oxygen O2(b)..." To read " particularly, by the emission of the Atmospheric Band which is produced by the emission of the excited state of molecular oxygen O2(b)..."

Lines39-40 Change "Lopez-Gonzales" to "Lopez-Gonzalez"

Line 66 Delete "O or"

Line 67 Change "by known" To read "and"

Line 67 add "values" to read " volume emission values"

Line 70 change "leads to the loss of self-consistency (e.g Murtagh et al., 1990)." To read "leads to some degree of uncertainty (e.g Murtagh et al., 1990)."

Line 70-71. Delete "and, consequently, to essential biases." To read "(e.g Murtagh et al., 1990)."

Line 73-74 Delete " real common volume in-situ measurements of these..." To read: "simultaneous measurements of these..."

Line 75-79 Change "chapter" To read "section"

2. Rocket experiment description

Here I have a question, it is described that FIPEX use two types of solid electrolyte sensors platinum electrodes sensitive to both molecular and atomic oxygen, whilst gold electrodes show a selective sensitivity to atomic oxygen. Is this mean that it can measure molecular oxygen too? If it is possible, It would be useful this O2 profile were shown in the plot.

3. Theory

All this section is just a very detailed recompilation of the known mathematical expressions used in O2b calculations. There are too many details that should be reduced.

I think expression (2) should be deleted is already said in line 135-136 (same that equation 1).

Then expression (4) and (5) should be deleted and put only expression (6). Then delete expression (7), it is not needed, and write expression (8).

Then (1),(3), (6) and (8) expressions could be written, to easy understand the "nomenclature".

4. Results and Discussion:

Comment: Figure 1, shows atmospheric concentration, [N], temperature, T, atomic oxygen concentration, [O], and Volume emission rate. Here Figures 1b) and 1c) should have additional subintervals in the logarithmic x axis. Additionally molecular oxygen concentration is used in the analysis performed. I have a question what values of O2 are used? (is there some additional measurement from FIPEX?, or are derived from the measurements of [N] (density) and [O]?)

Question: Lines 174-177. It said: "Our rocket experiment shows an essential difference of emissions between ascending and descending flights (see Strelnikov et al., 2018). It also demonstrates a significant variability in other measured parameters, including neutral temperature and density as well as atomic oxygen density."

How large is this difference? The time between ascending and descending flights should be a very few minutes. Perhaps it could be interesting to show the profiles of the different parameters obtained in both, the ascending and descending, flights to see these differences.

4.1 On-step mechanism.

This subsection has to be very simplified. There is a lot of speculative exercise that leads to any point.

For example: The efficiency calculated by using direct atomic oxygen recombination for the production of O2b to explain the observed emission, is in the range of about 0.07-0.13, this value is too large compare with the obtained by laboratory and theoretical investigation (Wraight, 1982; Ogryzlo et al., 1984; Bates, 1988). Then, direct atomic oxygen recombination alone can not explain the observed emission in agreement with earlier works (McDade et al., 1986; Bates, 1988;...)

Any other exercise is not necessary. So figure 2 y 3 should be deleted.

4.2 Two step mechanism.

Figure 4. Please, put some subintervals specially in the y axis. Here I see a very good fitted region of about 97-98 km (as I guess because of the lack of subintervals!), and important deviation of the fit above and below this region. I think it would be interesting to show, simultaneously in the figure, T profile (with a horizontal scale from 150K to 210K, in the upper x axis) and the N profile (the same scale as RHS Eq.8, in the lower x axis, is appropriated) to easy see how temperature and atmospheric density profiles affect to the features observed in RHS equation 8.

Line 259: Delete "s" to read " and molecular oxygen"

Line 261: Change "are 3.1 and 2.9" to read "of 3.1 and 2.9"

Line 282-283 Change "we show Figure 5 with atomic oxygen concentration from... " To read "we compare in Figure 5 the atomic oxygen concentration from ..."

((Figure 5 and Figure 7 need to show subintervals in both axes (vertical and horizontal), also the data points should be plotted each 1 km.))

4.3 Combined mechanism

Figure 6 is not necessary, all the information is in table 3.

I do not think to put the equation (9) is needed. Neither do I consider an appendix necessary. So I will deleted the appendix and will reduce the equation 9 and the explanations to: "... we have investigated a combined mechanism of direct and indirect atomic oxygen recombination, the fitting coefficients for the transfer energy process were calculated for.... The results for the best-fit in each case are listed in Table 3." (Now in Table 3 only K3o/k3o2 and total efficiency would shown (delete D1 and D2))

Line 305-310 Delete all.. "They are listed in Table 3. The altitude profile of the RHS of equation (9) and calculated fit-function are plotted in Figure 6. The deviations of fit function between limits and averaged values are negligible, hence, we only show the averaged case. Thus, we can recommend for future investigations the values of averaged case (last column of Tab. 3). Analogously to the two-step mechanism....<1.

Taking into..." To read: "The results for the best-fit in each case are listed in Table 3. Taking into..."

LIne 311. Add (see Table 3) to read "0.073 (see Table 3)."

Line 327-330. Delete this sentence "Note that... two point, respectively." It is not needed.

Line 330-332. Rewrite this sentence. For example: The total efficiency of production of O2b through an energy transfer process and new coefficients derived in this work provide a valuable information about the chemistry of O2b. Moreover, the importance of make studies with the possibility of using simultaneous measurements is strongly pointed out.

Conclusion:

This section has to be very summarised. There is a long text, and here the results has to be clearly established.

Line 336: Change "true common volume observations" to read "simultaneous observations"

Line 337-338 "delete one-step. two step and combined" to read "the mechanism of o2b formation were analysed".

The following discussion for one-step and two-step should be deleted. These are from line 339 to 356. Only the proposed mechanism should be mentioned.

Line 357-360 are when the main result are reflected. This can be rewritten as: Based on simultaneous observations of atomic oxygen, atmospheric band emission (762 nm), and density and temperature of the background atmosphere and all the information available up to now about reaction rates coefficients, branching ratios, quenching rates and spontaneous emission coefficients the mechanism of o2b formation were analysed.

A direct and indirect atomic oxygen recombination process to explain the production of O2(b) is the one chosen as responsible of the atmospheric emission observed. The total efficiency of production of O2b in the indirect recombination process is of 0.08 and the ratio of quenching coefficient of the precursor state is 0.231, when an efficiency of 0.02 in direct recombination is chosen. The analysis of the values of the total production indicates that O2A' or O2Pi may be possible precursors for the two-step mechanism.

The lines 361-366 reflect the final thoughts, so here Lines 361-366 ramble on about these mechanisms again in a confused way. Instead, it should show the need to make simultaneous measurements to confirm and improve these results.

---

## Referee Comment (RC2) · Anonymous Referee #3 · 12 Nov 2018

The paper "Atmospheric band fitting coefficients derived from self-consistent rocket-borne experiment" by Grygalashvyly et al uses observations of temperature, total air density, atomic oxygen and the $O_2(^1\Sigma)$ volume emission rate from a night-time rocket experiment of March 2015 to derive fitting coefficients for the formation of the emission signal considering three different formation pathways: direct formation, formation via an intermediate excited state, and a combination of both. The first two formation pathways are a repetition of a similar experiment from the ETON campaign as discussed in McDade et al 1976, which however had to use model values for temperature and total air density; the combination pathway is a new development as far as I know, though results suggest that the formation via an intermediate state dominates. This is an inter-

esting study, and considering that such fitting coefficients are used to derive night-time atomic oxygen density from observed volume emission rates of the O2 atmospheric band, highly relevant. However, I found that the paper clearly needs more work before final publication. My main concerns, listed below in more detail, concern the error analysis - errors are provided in some of the figures, but it is not explained how they are derived, and no error range is given for the end results of the analysis, the coefficients and efficiencies. This has to be provided in the final publication. Also, the derived coefficients are not compared directly to the results of the previous study by McDade et al; I found this really curious, as they are actually very different in particular concerning the coefficient of O-quenching, $C^O$. This really must be discussed. Can this large difference really be due to different temperature profiles used? And finally, the description of the data used, in particular of the volume emission rates, lacks important information needed to understand/interpret the results. In summary, I recommend publication only after major revisions, see list below.

**Major comments:**

Lines 107-111: considering that the volume emission rates are crucial for deriving the fitting coefficients for the $O(^3P)$-model, this explanation about how they are derived is totally inadequate. I appreciate that this might be explained in detail in Hedin et al. (2009), but information needed to interprete the results must be given here as well. These include: a) the viewing geometry of the instrument. Is it looking in flight direction of the rocket (in which case it would see roughly the same volume of air as the in-situ instruments at least when far away from the rockets apogee), or is it looking to the side (in which case it would not see the same volume of air as the in-situ instruments)? How are volume emission rates derived, and how large is the volume viewed by the instrument? How far away is it from the rocket path viewed by the in-situ instruments? What about the spectral resolution / coverage of the instrument? What's the measurement uncertainty, and why? What is the vertical resolution? Presumably

a few km, why? And how will that affect a comparison with in-situ observations which have a much better resolution?

Section 3: Throughout reading of section 3, I have wondered why you are not discussing a combined one-step/two-step mechanism. Turns out much later that you do, but the theory for that is discussed only in the Appendix. Why? As the combined mechanism seems more likely, and also seems to be a new development here, I would do it the other way round - discuss the combination here, and the individual mechanism in the Appendix. However, that is your decision (and I appreciate that you do not discuss all three here as that would be rather long). However, no matter whether you discuss the individual steps or the combined in section 3, you should really point out already in section three that you test all three posibilities (one-step, two-step, combined) in the paper, and that the derivation of the coefficients of the branches not discussed here is discussed in the Appendix.

Line 131-132: please also state that you have to assume photochemical equilibrium of $O_2(^1\Sigma)$ to derive equation (1).

Line 144: please also state that you have to assume photochemical equilibrium of O2* to derive equation (4).

Same in line 149: you have to assume photochemical equilibrium of $O_2(^1\Sigma)$. How valid is this assumption?

Lines 156-157: As you derive the coefficients $C^O$ and $C^{O2}$ over the whole altitude range, you have to assume that the coefficients are temperature independent. This means that the reaction rates $k_3$ are either temperature independent, or have the same temperature dependency for all quenching partners (N2, O2, O), correct? Is this a valid assumption?

Lines 194-196, Figure 2: What is the meaning of the error bars in Figure 2? How/from which uncertainties have they been derived? Considering the very large errors given

here, I would assume that this is not the measurement noise, as the error bars are actually much larger than the scatter of the measurement points.

Line 197: considering the error range of the individual altitudes, the range is rather 0.03-1.17. However, there is a common range including all data points and their errors which is much narrower, more like 0.07-0.10. However, you really should provide the most likely result based on the measurement statistics here, i.e., the mean or medium point plus/minus the standard error taking into account both the variance and the error range of the individual points.

Line 198: please provide the mean/median with error range based on the variance and error range of the individual points.

Line 199-200: considering the large error range, there is no significant altitude dependence. You can of course discuss it anyway, but please keep in mind (and state in the paper) that the variability of the data points is much smaller than the errors of the individual points.

Line 203: "values are distributed not randomly" ...well the altitude spacing shown in Figure 2 is obviously much smaller than the vertical resolution of the volume emission rates they are based on, compare to Figure 1 d), so I would expect of course there a non-random underlying altitude dependence - it comes from interpolation (or splining, or whatever) of the volume emission rate between data-points. Please discuss in the paper a) how volume emission rates are derived between data-points (interpolation, spline?), and b) how this affects the results.

Line 203: "clear functional dependence ..." well I see at least three functional dependencies here. I agree with your discussion below (lines 205-208) that one would expect a dependence on temperature and pressure due to the T/p dependence of the reaction rates; however, I think considering your large errors, and the fact that the low vertical resolution of the volume emission rates must imply an auto-correlation between data-points (see comments above), you can't really derive any evidence for this from your

data.

Line 235: values of $C^{O2}$ and $C^O$: please provide an error range based on error propagation from the error and variance of RHS as provided in Figure 4. Also, please compare these values directly with the values given by McDade et al 1976. Do they agree within the error range? The values are: your data: $[C^{O2}; C^O] = [9.8; 2.1]$; Mc-Dade: $[C^{O2}; C^O] = [4.8 - 7.5; 15 - 33]$. My expectation would be that they do not agree within your error range, particularly not $C^O$ which really is very different. Please discuss. Also, what could be the reason for the large discrepancy in $C^O$? Temperature dependence of the O-quenching? Or the use of the atomic oxygen profile? Please discuss.

Line 238: please discuss how well-funded the assumption is that quenching with N2 is much slower than quenching with O2. Is there any evidence for that?

Line 238: please provide error range for alpha gamma.

Line 250: please provide a symbol for the total efficiency, and use this symbol in the equation. tot.eff really looks unprofessional in the equation.

Line 257: please consider the error of alpha and the total efficiency 0.102 here. However, I would not expect this to change the conclusions here. Same in line 310.

Line 268: please provide error range

Lines 283 ff discussion of Fig 5: considering that you used the FIPEX data to constraints your coefficients, the agreement is not that good, actually; in particular, the shape of the profile appears slightly different, with the peak maximum at a higher altitude than the observation. In this, your result resemble the McDade results; maybe because in both cases, the ratio of two reaction rates is derived, not the rates themselves? In the lower part your results and those of McDade differ, presumably because $C^O$ is so different? Please discuss this difference from the comparison of the coefficients.

Line 316 ff discussion of Figure 7: Considering the volume emission rate observation you use here for comparison is the same that you used to constraint the coefficients of your model, this is more a sanity check than a validation. For a validation, you would have to compare your results to an independent measurement.

Line 349: please provide error range of results. Please make a statement about how those values compare to previous derivations (McDade 1976).

Lines 358, 359: please provide error range of results.

**Minor comments:**

Line 32-33: This sentence is too short; it's meaning is not clear at all. Please clarify: Why is the mesopause important for the upper atmosphere (which presumably is above the mesosphere?) Coupling between which atmospheric layers is important here? Presumably between mesosphere and thermosphere?

Line 40: "the tides parameters" is either "the tidal parameters" or "the tides' parameters"

Line 43: However, Takahashi et al used the (0,1) transition of the atmospheric band at 864,5 nm, while you use the (0,0) transition at 762 nm. The (0,0) transition was used however by Sheese et al CJP 2010, GRL 2011, and this should be discussed here.

Lines 54-55: please make clear that you are talking about the night-time population here; during day-time, the excitation mechanism of the atmospheric band are quite different, being dominated by $O(^1D)$ quenching and resonance fluorescence (see, e.g., Zarboo et al, AMT, 2018, Figure 9).

Lines 61-62: "the" hypothesis, "the" precursor

Lines 64-65: problem of identification is still not solved; the breakthrough ... this seems to be a contradiction. in particular considering that the ETON results were published in

1976 so are not recent at all. I agree that despite the ETON results there are still many open questions, but this should be formulated more carefully (and more clearly) here.

Line 126: "saving all nomenclature..." you mean "using all nomenclature..."?

Line 133: "fraction of recombination" this is actually the fraction of the three-body re-combination reaction that directly leads to $O_2(^1\Sigma)$. I found the term "fraction of recom-bination" misleading here (same in line 140); I would probably call this the "quantum yield of $O_2(^1\Sigma)$ formation".

Line 142-143: R8 is one pathway of the overall quenching reaction R9; this should be made clear here.

Line 227: "too low" really should be "much lower".

Line 233: It would help the reader to write "right hand side (RHS) of equation ..." once again here.

Line 233: "are amount to" $\longrightarrow$ "amount to"

Line 235: "in such a way define fitting-coefficient" $\longrightarrow$ "fitting-coefficients defined in such a way ..."

Line 320: erase one "the" before temperature.

Line 359: what does "a sense of" mean here?

Line 365: please erase "to" after "contradict"

Line 366: please insert "is" after "mechanism"

Figure 7: the figure is hard to read - lines are too thin, and the resolution appears to be low.

---

## Referee Comment (RC3) · Anonymous Referee #2 · 8 Dec 2018

General comments. The paper is an extension of the previous works related to the problem of atomic oxygen concentration derivation from nighttime observations of atmospheric exited O2 emission. In-situ consistent measurements of the temperature, air density, atomic oxygen concentration, and volume emission at 762 nm during WADIS-2 sounding rocket mission are used to correct the fitting coefficients for exited O2 emission parameterization. These corrected fitting coefficients may be useful to study dynamical and chemical processes in the mesosphere region. The advantage of these fitting coefficients is their self-consistence in contradiction to the previously derived ones. The paper may be recommended for publication after minor revision.

[Figure]

Specific comments. 1. It seems that there is an imbalance between the description of the three instruments used during the rocket mission. CONE and FIPEX are described in more or less detail, whereas for an Airglow Photometer only its functional purpose is mentioned. It is recommended to either shorten the description of the first two or expand the description of the Airglow Photometer. 2. It is not clear why the theory is divided into two parts. It seems that the Appendix can be combined with the "Theory" section, and one should begin with the first sentence of the Appendix about the assumption of photochemical equilibrium. In this case, it is desirable to discuss the possibility of using the assumption of photochemical equilibrium at night. In addition, despite the well-established term "photochemical equilibrium", for pure night conditions it is more correct to call it "chemical equilibrium". 3. It is necessary to describe the method of estimating the errors shown in the figures. With such large errors, it is necessary to speak of height dependence of fraction of recombination with caution. In addition, error estimates for the fitting coefficient estimates should also be presented. 4. More detailed comparison to the McDade et al. (1986) fitting coefficients is desirable taking into account error analysis.

Technical corrections. 1. Figure captures should be extended. 2. Figure 3 is not correlation. 3. Equations (1) and (A2) are the same. After combing theory section and appendix some equations may be omitted.
* * *

---

## Author Comment (AC1) · 18 Dec 2018

Dear Reviewer.

Thank you for taking a time to review our manuscript. We were trying to follow your suggestions.

In the following we address the comments of the reviewer point by point.

Reviewer write: "First, I do not consider necessary many of the analysis devoted to atomic oxygen direct recombination process. So Figures 2 and 3, and many of the discussion refereed to them, should not be here. Once the authors obtain from direct

recombination analysis that an efficiency of 0.07-0.13 is required for O2(b) production means that this process, alone, can not explain O2(b) production . . . ."

McDade et al. (1986) made the conclusion that one-step mechanism is not working not because their efficiency to high, but because it depends on altitude, that is impossible by the essence of $\varepsilon$. They are writing: "The altitude dependence of $\varepsilon$ suggests that the observations are not consistent with the direct excitation mechanism unless the efficiency for formation is strongly temperature dependent." Because our experiment is essentially differ (common volume observations) from former one, we should repeat the steps of McDade et al., (1986). Hence, figures 2, as well as short corresponding discussion should be saved. Please, approach with understanding. Nevertheless, in order to satisfy the reviewer suggestions, we delete figure 3 and radically reduce the discussion of one-step mechanism and made it as short as it is possible.

Reviewer write:" . . .then, all additional the work of considering only direct recombination process and possible dependences of the efficiency with temperature and pressure is very speculative, first because of the large value of this efficiency and second because in the altitude range of the atmospheric region considered ($\sim$10km) there is not large variation of temperature and pressure condition (although they could reach +-50K and/or a few micro bars)."

The temperature enters into Eq. (2) via coefficients , , under the exponential and power functions, hence, even small fluctuations of temperature produce essential deviations.

Reviewer write: "Figure 4, 5 and 7 need some improvements: 1) Plot some subintervals in the vertical and horizontal axes. 2) There are points (data) only at about each 3 km. Points at each 1 km should be shown, although the error bars be shown only each 3 km. 3) Figure 3 shows very good fit between about 97 and 98 km (as I can guess in figure 3 without any subdivision!). It will be interesting to show in the same figure the temperature profile (with and appropriate temperature scale in the upper horizontal axis), and the number density to see how the structure of temperature and number

density can affect the fitting."

We modify figures 4, 5 and 7 according your suggestions: 1) we plot some subintervals in the vertical and horizontal axes; 2) we prefer to use 3 km step at figures 4 and 5 (ex. Fig.5 and Fig. 7) because it is more appropriate to the volume emission observations (see Hedin et al., 2009); 3) we think that you mean figure 4, because the fit shown on figure 4, but not on figure 3. We bring our apologies that do not show in the same figure number density and temperature. This does not give any information how the structure of temperature and number density affects the fitting, because: there are three parameters O, T, M; eq. (4) non-linear, and the effects of these parameters not obvious; thus, we may not distinguish influences of different parameters. Moreover, this is not the subject of our paper and it led to defocusing.

Specific suggestions of the reviewer.

Abstract.

Reviewer write:" Line 19- 20: ",we derived the empirical fitting coefficients,....(0,0) in terms of the atomic oxygen concentrations." Delete "in terms of the atomic oxygen concentrations" To read: ",we derived the empirical fitting coefficients,....(0,0).""

Lines 19- 20 are changed according to Reviewer suggestion.

Line 25: Reviewer write:" Simultaneous and true common volume measurements of all the parameters used in this derivation, i.e...." Change to: Simultaneous measurements of all the parameters involved in the theoretical calculation of the observed O2(b) emission, i.e...."

"Common volume measurements" is a special term well known and generally accepted for in-situ observations. Hence, we should save this term. The rest of the sentence is modified according with Reviewer claim.

1. Introduction.

Reviewer write:" Line 33-35 Change: "particularly, by emissions in the Atmospheric Band that form the excited state of molecular oxygen O2(b)..." To read " particularly, by the emission of the Atmospheric Band which is produced by the emission of the excited state of molecular oxygen O2(b)..." Lines39-40 Change "Lopez-Gonzales" to "Lopez-Gonzalez" Line 66 Delete "O or" Line 67 Change "by known" To read "and" Line 67 add "values" to read " volume emission values" Line 70 change "leads to the loss of self-consistency (e.g Murtagh et al., 1990)." To read "leads to some degree of uncertainty (e.g Murtagh et al., 1990)." Line 70-71. Delete "and, consequently, to essential biases." To read "(e.g Murtagh et al., 1990)." Line 73-74 Delete " real common volume in-situ measurements of these..." To read: "simultaneous measurements of these..." Line 75-79 Change "chapter" To read "section""

Lines 33-35 are changed according to Reviewer suggestion. Lines 39-40 are corrected as Reviewer suggest. Lines: 66-67. The sentence is rewritten more clearly. Line: 67. The word "values" is added after the words "volume emission" as Reviewer suggest. Line 70 is modified according with Reviewer suggestion. Lines 70-71 are changed deleted by Reviewer suggestion. Line: 74. We save "common volume" as special term (see answer above), but the rest the sentence is modified according with Reviewer wish. Lines: 75-79. The word "chapter" is changed to "section" through the entire manuscript.

2. Rocket experiment description.

Reviewer write: "Here I have a question, it is described that FIPEX use two types of solid electrolyte sensors platinum electrodes sensitive to both molecular and atomic oxygen, whilst gold electrodes show a selective sensitivity to atomic oxygen. Is this mean that it can measure molecular oxygen too? If it is possible, It would be useful this O2 profile were shown in the plot."

We use the CONE number density measurements and partial partitioning from NRLMSISE-00 reference atmosphere (Picone et al., 2002). The FIPEX O2 density

measurements are not used: for reasoning and discussion see Eberhart et al. (2015). To avoid such confusion we rewrote the FIPEX experiment description.

3. Theory.

Reviewer write: "All this section is just a very detailed recompilation of the known mathematical expressions used in O2b calculations. There are too many details that should be reduced. I think expression (2) should be deleted is already said in line 135-136 (same that equation 1). Then expression (4) and (5) should be deleted and put only expression (6). Then delete expression (7), it is not needed, and write expression (8). Then (1),(3), (6) and (8) expressions could be written, to easy understand the "nomenclature"."

All of Reviewer suggestions are applied. Equations (2), (4), (5), and (7) are deleted.

4. Results and Discussion.

Reviewer write: "Comment: Figure 1, shows atmospheric concentration, [N], temperature, T, atomic oxygen concentration, [O], and Volume emission rate. Here Figures 1b) and 1c) should have additional subintervals in the logarithmic x axis. Additionally molecular oxygen concentration is used in the analysis performed. I have a question what values of O2 are used? (is there some additional measurement from FIPEX?, or are derived from the measurements of [N] (density) and [O]?)"

We add subintervals in Figure 1. Molecular oxygen is derived from CONE atmospheric number density measurements and partitioning from NRLMSISE-00 reference atmosphere (Picone et al., 2002). We add the notation into the section 2.

Reviewer write: "Question: Lines 174-177. It said: "Our rocket experiment shows an essential difference of emissions between ascending and descending flights (see Strelnikov et al., 2018). It also demonstrates a significant variability in other measured parameters, including neutral temperature and density as well as atomic oxygen density." How large is this difference? The time between ascending and descending flights

should be a very few minutes. Perhaps it could be interesting to show the profiles of the different parameters obtained in both, the ascending and descending, flights to see these differences."

We add the references where such difference is shown and discussed. Following Reviewer advice that the paper should have better focus we do not repeat this discussion in our manuscript.

4.1 One-step mechanism.

Reviewer write: "This subsection has to be very simplified. There is a lot of speculative exercise that leads to any point."

Situations in which the science allows a formulation of several reasonable alternatives, and it is impossible to show convincingly that only some one of them is right, are characteristic of all fields of scientific research. In this case researcher should discuss all available alternatives.

Reviewer write: "For example: The efficiency calculated by using direct atomic oxygen recombination for the production of O2b to explain the observed emission, is in the range of about 0.07-0.13, this value is too large compare with the obtained by laboratory and theoretical investigation (Wraight, 1982; Ogryzlo et al., 1984; Bates, 1988). Then, direct atomic oxygen recombination alone can not explain the observed emission in agreement with earlier works (McDade et al., 1986; Bates, 1988;...) Any other exercise is not necessary. So figure 2 y 3 should be deleted.".

As we discuss above, the figure 2 is saved but the section essentially simplified and figure 3 is deleted.

4.2 Two-step mechanism.

Reviewer write:" Figure 4. Please, put some subintervals specially in the y axis. Here I see a very good fitted region of about 97-98 km (as I guess because of the lack of subintervals!), and important deviation of the fit above and below this region. I

think it would be interesting to show, simultaneously in the figure, T profile (with a horizontal scale from 150K to 210K, in the upper x axis) and the N profile (the same scale as RHS Eq.8, in the lower x axis, is appropriated) to easy see how temperature and atmospheric density profiles affect to the features observed in RHS equation 8."

To address the Reviewer's point we put subintervals in figure 4. T and M on figure 4 does not give an ability to see how the temperature and number density affects the RHS, because beside T and M there is third parameter (O) and one non-linear eq. (4) from which we may not distinguish influences of different parameters. Moreover, this is not the subject of our paper and it led to defocusing.

Reviewer write: "Line 259: Delete "s" to read " and molecular oxygen" Line 261: Change "are 3.1 and 2.9" to read "of 3.1 and 2.9" Line 282-283 Change "we show Figure 5 with atomic oxygen concentration from... " To read "we compare in Figure 5 the atomic oxygen concentration from ..." ((Figure 5 and Figure 7 need to show subintervals in both axes (vertical and horizontal), also the data points should be plotted each 1 km.))"

Line: 259. "s" is deleted. Line: 261. Changed. Line: 282-283. Changed. The subintervals in both axes at Figure 5 and Figure 7 (now Fig. 4 and Fig. 5) are added. We save resolutions 3 km as it is more relevant to volume emission observations.

4.3 Combined mechanism

Reviewer write:" Figure 6 is not necessary, all the information is in table 3."

This contradicts to (*). Nevertheless, in order to satisfy the Reviewer suggestion we delete Figure 6.

Reviewer write: "I do not think to put the equation (9) is needed. Neither do I consider an appendix necessary. So I will deleted the appendix and will reduce the equation 9 and the explanations to: "... we have investigated a combined mechanism of direct and indirect atomic oxygen recombination, the fitting coefficients for the transfer energy

process were calculated for.... The results for the best-fit in each case are listed in Table 3." (Now in Table 3 only K3o/k3o2 and total efficiency would shown (delete D1 and D2)) "

We need this equation because it used to calculate fitting coefficients, hence, we have to show how it was derived, consequently, we save an appendix.

Reviewer write: "Line 305-310 Delete all.. "They are listed in Table 3. The altitude profile of the RHS of equation (9) and calculated fit-function are plotted in Figure 6. The deviations of fit function between limits and averaged values are negligible, hence, we only show the averaged case. Thus, we can recommend for future investigations the values of averaged case (last column of Tab. 3). Analogously to the two-step mechanism....<1.Taking into..." To read: "The results for the best-fit in each case are listed in Table 3. Taking into..." "

This contradicts to (*) and (**). Nevertheless, in order to satisfy the suggestion of the Reviewer we delete lines 305-310 and add "The results for the best-fit in each case are listed in Table 3."

Reviewer write:" LIne 311. Add (see Table 3) to read "0.073 (see Table 3).""

We add "(see Tab. 3)".

Reviewer write:" Line 327-330. Delete this sentence "Note that... two point, respectively." It is not needed."

This contradicts to (*). Nevertheless, in order to satisfy the suggestion of the Reviewer the sentences at lines 327-330 have been deleted.

Reviewer write:" Line 330-332. Rewrite this sentence. For example: The total efficiency of production of O2b through an energy transfer process and new coefficients derived in this work provide a valuable information about the chemistry of O2b. Moreover, the importance of make studies with the possibility of using simultaneous measurements is strongly pointed out."

This contradicts to (*) and (**). Nevertheless, in order to satisfy the Reviewer the sentences were rewritten.

Conclusion.

Reviewer write: "This section has to be very summarised. There is a long text, and here the results has to be clearly established."

Following by you suggestion the text of conclusion has been reduced.

Reviewer write: "Line 336: Change "true common volume observations" to read "simultaneous observations". "

"Common volume" is special term (see answer above). Trying to follow your suggestion we change "true common volume observations" to "common volume simultaneous observations".

Reviewer write: "Line 337-338 "delete one-step. two step and combined" to read "the mechanism of o2b formation were analysed"." We delete "one-step. two step and combined" to "the mechanisms of formation were analysed."

Reviewer write: "The following discussion for one-step and two-step should be deleted. These are from line 339 to 356. Only the proposed mechanism should be mentioned."

This contradicts to (*) and (**). In this paper we discus three mechanisms, hence, in conclusion should be highlighted all of them. Nevertheless, in order to satisfy the suggestion of the Reviewer we essentially reduce the conclusion owing to one-step mechanism.

Reviewer write: "Line 357-360 are when the main result are reflected. This can be rewritten as: Based on simultaneous observations of atomic oxygen, atmospheric band emission (762 nm), and density and temperature of the background atmosphere and all the information available up to now about reaction rates coefficients, branching ratios, quenching rates and spontaneous emission coefficients the mechanism of o2b

formation were analysed. A direct and indirect atomic oxygen recombination process to explain the production of O2(b) is the one chosen as responsible of the atmospheric emission observed. The total efficiency of production of O2b in the indirect recombination process is of 0.08 and the ratio of quenching coefficient of the precursor state is 0.231, when an efficiency of 0.02 in direct recombination is chosen. The analysis of the values of the total production indicates that O2A' or O2Pi may be possible precursors for the two-step mechanism."

We do not include the reviewer text instead of our conclusions because this contradicts to (*).

Reviewer write: "The lines 361-366 reflect the final thoughts, so here Lines 361-366 ramble on about these mechanisms again in a confused way. Instead, it should show the need to make simultaneous measurements to confirm and improve these results."

Note, lines 361-366 were written by authors. The Reviewer term "ramble on about" unacceptable. This contradicts to (*) and (**), hence, we save this formulation as it is. Nevertheless, in order to satisfy the suggestion of the Reviewer we include at the end of the conclusion the statement of the Reviewer about more simultaneous measurements in future to confirm and improve these results.

Thank you again.

With respect, M. Grygalashvyly, M. Eberhart, J. Hedin, B. Strelnikov, F.-J. Lübken, M. Rapp, S. Löhle, S. Fasoulas, M. Khaplanov, J. Gumbel, and E. Vorobeva.

(*) 3. A referee of a manuscript should judge objectively the quality of the manuscript and respect the intellectual independence of the authors. In no case is personal criticism appropriate. (**) 7. Referees should explain and support their judgements adequately so that editors and authors may understand the basis of their comments.

(General obligations for referees, ACP web page, https://www.atmospheric-chemistry-and-physics.net/for_reviewers/obligations_for_referees.html)

---

## Author Comment (AC2) · 18 Dec 2018

Dear Referee,

Thank you very much for your constructive suggestions. We tried to follow your comments and suggestions, but, please, approach with an understanding, that we should search a compromise between your suggestions and suggestions of other reviewers.

Major comments.

It is true that it is necessary to better inform potential readers about volume emission measurements. Now we add such description at lines 115-135 of the revised

manuscript, where, we hope, all of your questions are highlighted.

Section 3.

We pointed out now directly in Section 3 that we consider combined mechanism in section 4.3 and derive an expression for corresponding fit-function in Appendix (lines 190-192 of the revised manuscript). Now we mention in Sec. 3 that the assumptions about photochemical equilibrium for O2(b1$\Sigma$) and O2* are used (lines 155-156, 171-172, and 174-178 of the revised manuscript) . We add the notation that the coefficients CO and CO2, and consequently k3, are assumed to be temperature independent (or dependence is weak) and short discussion of this assumption (lines 183-188 of the revised manuscript).

Line 194-196 (hereafter, line numbers at the beginning as in review), Figure 2. The uncertainties were calculated according with sensitivity analysis. Now we mention this directly in the manuscript and give the references (lines 231-233 of the revised manuscript). The sensitivity analysis allows to estimates uncertainty of target component on the basis of errors of parameters of given component. Advantage of the sensitivity analysis consists that it considers contribution of each parameter to uncertainty of target component at the expense of sensitivity coefficients. In our case, the dependence of target component on parameters is known as well as the errors of these parameters. Thus, calculating all sensitivity coefficients (partial derivatives of target components for each parameter), we define the resulting uncertainty of each target component.

Line 197. Information about $\varepsilon$ taking into account both the variance and the error range has been included.

Line 198. The information about mean with error range based on the variance and error range of the individual points is provided.

Line 199-200. The corresponding discussion is corrected. Now we mark that considering the large error range, there is no significant altitude dependence and state in the paper that the variability of the data points is much smaller than the errors of the individual points (lines 238-241 of the revised manuscript).

Line 203. This is true. Considering our large errors, we can't really derive any evidence for functional dependence. By demand of 1st reviewer the figure 3 has been deleted, as well as corresponding discussion.

Line 235. We provide an error range based on error propagation from the error and variance of RHS as provided in Figure 3. Now we compare these values directly with the values given by McDade et al. (1986). Possible reasons for the large discrepancy in CO are noted (lines 254-264 of the revised manuscript).

Line 238. We add the notation that the assumption (K3N2«K3O2) is just working hypothesis which is commonly used for analysis of precursor and, currently, there is no any evidence, neither for nor against that. If it is not true any definite conclusion on precursor by known CO2 is not possible (lines 267-273 of the revised manuscript).

Line 238. The error range for $\alpha\gamma$ has been provided (line 273 of the revised manuscript).

Line 250. We provides $\eta=\alpha\gamma$ as symbol for total efficiency for two-step mechanism and analogously total efficiency for two-step channel at combined mechanism at line 274 of the revised manuscript.

Line 257. We add consideration of uncertainty for total efficiency. At lower limit of uncertainty the result is saved, and considering upper limit, only O2(5Pi) may serve as precursor (lines 292-294 of the revised manuscript).

Line 268. We provided an error range for CO/CO2 ratio (line 305 of the revised manuscript).

Line 283. We add recommended discussion (lines 327-333 of the revised manuscript).

[Figure]

Line 316. Now we note directly in the paper that Figure 7 (now Figure 5) is rather a sanity check than a validation (line 360 of the revised manuscript). Unfortunately we do not have other independent observation in present time. We add a notation about the necessity more independent common volume in-situ measurements to validate this results (lines 372-373 of the revised manuscript).

Line 349. We add error range and made a statement how our values compare to previous of McDade et al. (1986) (lines 383-386 of the revised manuscript).

Line 358, 359. Now we provide the error range for these values (line 394 of the revised manuscript).

Minor comments.

All of your minor suggestions were utilized. The text at lines 32-33 has been corrected. The lines 64-65 were reformulated more carefully. We studied this work now and add into the reference list.

Other changes are related to the recommendations and demands of other referee. Thank you a lot for taking your time to review our manuscript.

With respect, M. Grygalashvyly, M. Eberhart, J. Hedin, B. Strelnikov, F.-J. Lübken, M. Rapp, S. Löhle, S. Fasoulas, M. Khaplanov, J. Gumbel, and E. Vorobeva.

---

## Author Comment (AC3) · 18 Dec 2018

Dear Referee,

Thank you a lot for your constructive suggestions. We tried to follow your comments and suggestions. Please, approach with an understanding, that we should search a compromise between your suggestions and suggestions of other reviewers.

Specific comments.

Reviewer write: "1. It seems that there is an imbalance between the description of the three instruments used during the rocket mission. CONE and FIPEX are described

in more or less detail, whereas for an Airglow Photometer only its functional purpose is mentioned. It is recommended to either shorten the description of the first two or expand the description of the Airglow Photometer."

We extend the description of the airglow photometer (lines 115-135 of the revised manuscript).

Reviewer write: "2. It is not clear why the theory is divided into two parts. It seems that the Appendix can be combined with the "Theory" section, and one should begin with the first sentence of the Appendix about the assumption of photochemical equilibrium. In this case, it is desirable to discuss the possibility of using the assumption of photochemical equilibrium at night. In addition, despite the well-established term "photochemical equilibrium", for pure night conditions it is more correct to call it "chemical equilibrium"."

We decided move derivation of equation for combined mechanism into Appendix in order to make the paper shorter and better focused. Now we discuss assumption about photochemical equilibrium at nigh directly in the section "Theory" (lines 174-178 of the revised manuscript). We use term photochemical equilibrium because it is more general and because even at night conditions exited molecular oxygen is a subject of sponateouse emission.

Reviewer write: "3. It is necessary to describe the method of estimating the errors shown in the figures. With such large errors, it is necessary to speak of height dependence of fraction of recombination with caution. In addition, error estimates for the fitting coefficient estimates should also be presented."

The uncertainties were calculated with sensitivity analysis. We add some references where full description of this method is given (lines 231-233 of the revised manuscript). The discussion of Figure 2 and hight dependence has been modified taking into account large error (lines 238-241 of the revised manuscript). The uncertainties of the fitting coefficients for two-step mechanism, as well as for combined mechanism fitting

coefficients are shown in the revised manuscript.

Reviewer write: "4. More detailed comparison to the McDade et al. (1986) fitting coefficients is desirable taking into account error analysis."

It is true that it is necessary to better inform potential readers about comparison with coefficients of McDade et al. (1986). Now we add such discussion at lines 254-264 of the revised manuscript (taking into account uncertainties), where, we hope, this subject is highlighted.

Technical corrections.

Reviewer write: "1. Figure captures should be extended. 2. Figure 3 is not correlation. 3. Equations (1) and (A2) are the same. After combing theory section and appendix some equations may be omitted."

All of your technical corrections were utilized. 1. The captions for all figures are extended. 2. Figure 3 has been deleted by the suggestion of Reviewer 1. 3. Here, we think you mean Eq. 4 and (A2). By the suggestions of Reviewer 1 Eq. 4 (as well as Eq. 2, 5, 7) has been deleted.

Other changes are related to the recommendations and demands of other referee. Thank you a lot for taking the time to review our manuscript.

With respect, M. Grygalashvyly, M. Eberhart, J. Hedin, B. Strelnikov, F.-J. Lübken, M. Rapp, S. Löhle, S. Fasoulas, M. Khaplanov, J. Gumbel, and E. Vorobeva.
* * *

---

## Author Response (AR2)

Third response to the comments of the Reviewer 1

**Atmospheric Band Fitting Coefficients Derived from Self-Consistent Rocket-Borne Experiment.**

By M. Grygalashvyly, M. Eberhart, J. Hedin, B. Strelnikov, F.-J. Lübken, M. Rapp, S. Löhle,

S. Fasoulas, M. Khaplanov, J. Gumbel, and E. Vorobeva

Dear Referee,

Thank you very much for your positive mark of our work and constructive corrections. Almost all of your corrections were utilized.

Page 7. Line 155. Change "as a main one" to read "as the main one".

Changed.

Page 7. Line 161. I suggest to add and combine here lines 164-165 to read something similar to: "Then the volume emission, Vat, is obtained multiplying the O2b concentration by the spontaneous emission coefficient of the (0-0) band, A1, of reaction R5 (hereafter, nomenclature RX means the reaction X for table 1)".

We add such formulation at Line 161.

Page 10. Line 235. I suggest to change "0.11+/- 0.018" to read "011+/-0.02".

Changed.

Page. 14. Line 331. Add "the" to read: "is larger than the term with".

We add "the".

Page. 14. In my opinion the expression 5 can be introduced in line 353, by explaining that is the expression derived from combined mechanism (expressions 1 and 3) that similar to two step mechanism (is rearranged in expression 4) can be rearranged as: (Then, write expression 5). In my opinion I do not find necessary to repeat these expressions in an appendix.

We decided leave this place as it is. Please, approach with an understanding.

Page. 16. Line 380. Add ",alone," to read. "..direct excitation, alone, is less probable..."

Changed.

Page 16. Lines 395-396. Add the values "0.08" and "0.231" to read: "... of total efficiency alphagamma= 0.08 and a ratio of kO/KO2=0.231 for the two-step..."

It is corrected according with your suggestion.

Thank you a lot for taking your time to review our manuscript and for your comprehensive and unformal approach to our work.

With respect,

M. Grygalashvyly, M. Eberhart, J. Hedin, B. Strelnikov, F.-J. Lübken, M. Rapp, S. Löhle, S. Fasoulas, M. Khaplanov, J. Gumbel, and E. Vorobeva.

Second Response to the comments of the Reviewer 3

**Atmospheric Band Fitting Coefficients Derived from Self-Consistent Rocket-Borne**

                                **Experiment.**

By M. Grygalashvyly, M. Eberhart, J. Hedin, B. Strelnikov, F.-J. Lübken, M. Rapp, S. Löhle,

S. Fasoulas, M. Khaplanov, J. Gumbel, and E. Vorobeva

Dear Referee,

Thank you very much for your positive mark of our work and constructive technical corrections and suggestions. All of your corrections were utilized.

Line 538: write out acronyms (NLC, PMSE) once.

We add decipherment for NLC and PMSE.

Line 574: The essential step ... has been "made"

Changed.

Line 591: Capitals or not capitals? Use consistently

Changed.

Line 662-663: the sentence misses a verb, maybe "is in photochemical equilibrium"?

Corrected. We add "is".

Line 705-706: "is submitted true" should be "is assumed true"?

Yes, it should be "is assumed true". It is corrected.

Line 709: "such ability" better "such processes" or "the ability for such processes"?

We change this formulation. Now we write "such processes".

Line 750 and following: please just state what you used - error propagation from the precursors, variance, fitting errors?

We add such statements.

Other changes are related to the corrections of other referee. Thank you a lot for taking your time to review our manuscript and for your comprehensive and not formal approach to our work.

With respect,

M. Grygalashvyly, M. Eberhart, J. Hedin, B. Strelnikov, F.-J. Lübken, M. Rapp, S. Löhle, S.

Fasoulas, M. Khaplanov, J. Gumbel, and E. Vorobeva.

[revised manuscript text omitted]